# Reconstructed global monthly burned area maps from 1901 to 2020

Zhixuan Guo[1], Wei Li[1,*], Philippe Ciais[2], Stephen Sitch[3], Guido R. van der Werf[4], Simon P. K. Bowring[2], Ana Bastos[5], Florent Mouillot[6], Jiaying He[7], Minxuan Sun[1], Lei Zhu[1], Xiaomeng Du[1], Nan Wang[1], Xiaomeng Huang[1]

[1]Department of Earth System Science, Ministry of Education Key Laboratory for Earth System Modeling, Institute for Global Change Studies, Tsinghua University, Beijing 100084, China
[2]Laboratoire des Sciences du Climat et de l'Environnement, LSCE/IPSL, CEA-CNRS-UVSQ, Université Paris-Saclay, 91191 Gif-sur-Yvette, France
[3]Faculty of Environment, Science and Economy, University of Exeter, Exeter, EX4 4QF, UK
[4]Environmental Sciences Group, Wageningen University, P.O. Box 47, 6700AA, Wageningen, the Netherlands
[5]Land-Atmosphere Interactions, Institute Building, Talstraße 35, Room 2-16, 04103, Leipzig, German
[6]UMR CEFE 5175, CNRS, Université de Montpellier, Université Paul-Valéry Montpellier, EPHE, IRD, 1919 route de Mende, 34293 Montpellier CEDEX 5, France
[7]State Key Laboratory of Resources and Environmental Information System, Institute of Geographic Sciences and Natural Resources Research, Chinese Academy of Sciences, Beijing 100101, China

*Correspondence to*: Wei Li (wli2019@tsinghua.edu.cn)

**Abstract.** Fire is a key Earth System process, driving variability in the global carbon cycle through $CO_2$ emissions into the atmosphere and subsequent $CO_2$ uptake through vegetation recovery after fires. Global spatiotemporally consistent datasets on burned area are available since the beginning of the satellite era in the 1980s but are sparse prior to that date. In this study, we reconstructed global monthly burned area at a resolution of $0.5°×0.5°$ from 1901 to 2020 using machine learning models trained against satellite observed burned area between 2003 and 2020, with the goal of reconstructing long-term burned area information to constrain historical fire simulations. We first conducted a classification model to separate grid cells with extreme (burned area $\geq$ the 90th percentile in a given region) and regular fires, and then trained separate regression models for grid cells with extreme or regular fires. Both the classification and regression models were trained against a satellite-based burned area product (FireCCI51) based on explanatory variables related to climate, vegetation, and human activities. The trained models can well reproduce the long-term spatial patterns (slopes = 0.70-1.28 and $R^2$ = 0.69-0.98 spatially), inter-annual variability and seasonality of the satellite-based burned area observations. After applying the trained model to the historical period, the predicted annual global total burned area ranges from 3.46 to 4.58 million $km^2$ $yr^{-1}$ (M $km^2$ $yr^{-1}$) over 1901-2020 with regular and extreme fires accounting for 1.36-1.74 and 2.00-3.03 M $km^2$ $yr^{-1}$ respectively. Our models estimate a global decrease in burned area during 1901-1978 (slope = -0.009 M $km^2$ $yr^{-2}$), followed by an increase during 1978-2008 (slope = 0.020 M $km^2$ $yr^{-2}$) and then a stronger decline in 2008-2020 (slope = -0.049 M $km^2$ $yr^{-2}$). Africa was the continent with largest burned area globally during 1901-2020, and its trends also dominated the global trends. We validated our predictions against charcoal records, and our product exhibits a high overall accuracy in fire occurrence (>80%) in boreal North America, southern Europe, South America, Africa and southeast Australia, but the overall accuracy is relatively lower in northern Europe and Asia (<50%). In addition, we compared our burned area data with multiple independent regional

burned area maps in Canada, USA, Brazil, Chile and Europe, and found general consistency in the spatial patterns (linear regression slopes ranging 0.84-1.38 spatially) and the inter-annual variability. The global monthly 0.5°×0.5° burned area fraction maps from 1901 to 2020 presented by this study can be freely downloaded from https://doi.org/10.5281/zenodo.14191467 (Guo and Li, 2024).

## 1 Introduction

Fire is an important component of the Earth System (Bowman et al., 2009; Bowman et al., 2020) with large impacts on ecosystems by altering vegetation structure and function (Bond et al., 2005; Lasslop et al., 2016) and affecting the regional or even global energy budget through changes in surface albedo (Randerson et al., 2006) and emissions of aerosols and greenhouse gases (Van Der Werf et al., 2010). Vegetation recovery after fires also contributes to a legacy carbon flux to the ecosystem carbon sink (Hudiburg et al., 2023; Song et al., 2018; Yue et al., 2020). On the other hand, fire occurrence and spread are controlled by complex factors such as climate conditions, vegetation states, ignition foci, anthropogenic activities and their interactions (Andela et al., 2017; Flannigan et al., 2009; Jones et al., 2022; Senande-Rivera et al., 2022). Therefore, accurately mapping the spatiotemporal patterns of global burned area is essential for understanding mechanisms of fire disturbances, quantifying the global carbon budget and local energy balance (Mouillot et al., 2014).

Satellites provide direct observations of fire activities (e.g., burned area, fire radiative power) (Andela et al., 2017; Giglio et al., 2006; Luo et al., 2024), but they have limited temporal coverages because most satellite data were available only after 1980s (Chuvieco et al., 2019). Fire modules in dynamic global vegetation models (DGVMs) are able to simulate long-term burned area and the interactions with vegetation dynamics based on climate conditions and soil properties (Sitch et al., 2015; Sitch et al., 2024), but the spatial resolution at the global scale is usually coarse due to the coarse resolution of the input meteorological forcing data, and most models failed in capturing the global trends of burned area (Andela et al., 2017; Hantson et al., 2020). The processes included and the parameterizations of fire processes are widely different across fire models, resulting in a large range of simulated burned area at both the regional and global scales (Hantson et al., 2020). Considering the limitations of satellite observations and fire models, spatiotemporally consistent burned area maps over the 20[th] century trained from present-day observations, are essential for fire modelling and can serve as publicly available benchmark for fire ecology and carbon cycle studies.

A previous study synthesized historical statistics of burned area and reconstructed the global fire decadal history in the 20[th] century based on statistical models and a prescribed fire probability map from satellites (Mouillot and Field, 2005). This dataset is valuable since it incorporates many historical national fire records, despite being prone to uncertainties in the extrapolation. Machine learning models are now widespread and constitute appropriate tools to capture the non-linearity in complex systems such as wildfire, and have been used to predict burned area based on climate, fuel conditions and anthropogenic activities, but the temporal coverage of predictions is sometimes limited by the input data (Jain et al., 2020; Joshi and Sukumar, 2021; Li et al., 2023). There have been attempts to integrate machine learning models to replace process-

based wildfire models in Earth System models, and machine learning models exhibit better performance than process-based wildfire models even though they highly depend on the input data simulated with uncertainty from earth system models (Zhu et al., 2022). It is challenging for machine learning models to predict extreme values because extreme values are often treated as outliers and limited by the sample size (Breunig et al., 2000; Ribeiro and Moniz, 2020). However, extreme fires, usually defined as fires with an unprecedented scale or intensity (Bowman et al., 2017; Castro Rego et al., 2021; Cunningham et al., 2024), have significantly greater impacts than regular fires by releasing more $CO_2$, altering hydrological cycles, and emitting higher levels of pollutants (Clarke et al., 2022; Page et al., 2011).

In this study, we produced a global monthly $0.5°\times0.5°$ burned area fraction (BAF) dataset from1901 to 2020 (Guo and Li, 2024) using machine learning models based on climate conditions, vegetation states, population density and land use data (Table 1). To better capture extreme fires, we first developed a classification model to distinguish grid cells with extreme and regular fires, using the 90th percentile of all burned area fractions within a region as the threshold to define extreme fires. We then trained separate regression models for grid cells categorized as having extreme or regular fires. The models were trained against the satellite-based burned area product (FireCCI51) during 2003-2020, and then used to reconstruct the burned area from 1901 to 2020. In addition to evaluation against satellite observations that were not used for model training, we also compared our burned area predictions with charcoal records and other independent global and regional burned area datasets (Table 2).

## 2 Methods

The workflow of this study is illustrated in Fig. 1. The datasets used for extracting predictors and comparisons are listed in Table 1 and Table 2, respectively. We first divided the globe into 14 regions (Fig. S1) following the Global Fire Emission Dataset (GFED regions) (Giglio et al., 2006; Van Der Werf et al., 2017) and conducted machine learning model training, testing and prediction in each GFED region individually. The 14 GFED regions were abbreviated as BONA (Boreal North America), TENA (Temperate North America), CEAM (Central America), NHSA (Northern Hemisphere South America), SHSA (Southern Hemisphere South America), EURO (Europe), MIDE (Middle East), NHAF (Northern Hemisphere Africa), SHAF (Southern Hemisphere Africa), BOAS (Boreal Asia), CEAS (Central Asia), SEAS (Southeast Asia), EQAS (Equatorial Asia) and AUST (Australia and New Zealand).

### 2.1 Data preparation

We used the satellite-based monthly global burned area grid product FireCCI51 from 2003 to 2020 (Lizundia-Loiola et al., 2020) for the model training. This dataset was resampled from the original resolution $0.25°\times0.25°$ to $0.5°\times0.5°$. We excluded all burned pixels overlapping cropland classes in the CCI land-cover layer provided with FireCCI51 (Lizundia-Loiola et al., 2020) to remove agricultural fires from our analysis. We used the 90th percentile of all burned area fractions in $0.5°\times0.5°$ grid cells within a region as the threshold to define extreme fires. This percentile was chosen based on previous literature

(Bowman et al., 2017; Cunningham et al., 2024; Lannom et al., 2014). It is high enough ($\geq 90^{th}$) to distinguish moderate from extreme samples to train separate models for each category. Meanwhile, it is not too high (e.g., $95^{th}$ or $99^{th}$) in regions with limited data (such as Europe and the Middle East) to ensure sufficient extreme samples for model training and evaluation. The monthly distribution of burned area within $0.5°\times0.5°$ grid cells (Fig. S2) shows that if regular and extreme fires are modeled together (black curves), the abundant moderate values drown out the extremes (orange curves), causing total area to be underestimated. We thus first conducted classification and then trained separate models for regular and extreme burned fractions to enhance the representation of extreme events and improve regression performance. We used 16 explanatory variables to represent climate, vegetation and anthropogenic effects on regular and extreme BAF in machine learning models (Table 1).

Climatic variables including daily maximum temperature (Tmax), daily minimum temperature (Tmin), precipitation (Precip) and wind speed (Wind) were directly extracted and resampled to the monthly time step from CRUJRA v2.2, a global climate forcing dataset covering a time span from 1901 to 2020 with 6-hourly temporal resolution and $0.5°\times0.5°$ spatial resolution (Harris et al., 2014; Harris et al., 2020; Kobayashi et al., 2015). Vapor pressure deficit (VPD) was calculated based on empirical equations (Buck, 1981) using air temperature, air pressure and specific humidity from CRUJRA v2.2 at the 6-hour step and then averaged to monthly. Fire weather index (FWI), a numeric rating of fire intensity in Canadian Forest Fire Weather Index System (Wagner, 1987), was calculated using the 'cffdrs' package (Wang et al., 2017) in R programming language (R, 2024) with air temperature, relative humidity, wind speed and precipitation from CRUJRA v2.2 at the daily step and then averaged to monthly.

Variables related to anthropogenic effects include population density, land use and land use change fractions. Population density was resampled to $0.5°\times0.5°$ from HYDE3.2, a global population density dataset with a spatial resolution of 5 arc minutes available from 10,000 before Common Era to 2015 Common Era (Klein Goldewijk et al., 2017). Land use fractions refer to the area fraction of a certain land use type in each $0.5°\times0.5°$ grid cell in the current year, and land use change fractions are the difference of land use fractions between the current year and the previous year. We use the area fractions of four land use types (forest, shrub, natural grass and cropland) from the ESA CCI land cover maps during 1992-2020 (Li et al., 2018), and the Land Use Harmonization 2 dataset for Global Carbon Budget 2020 (LUH2-GCB2020) for the period before 1992 (Chini et al., 2021), following the methods by (Peng et al., 2017). In the land use harmonization process, we resampled both ESA CCI and LUH2 datasets to $0.5°\times0.5°$ and reclassified them into five land use types (forest, shrub, natural grass, cropland and others). The above five land use types were converted from the ESA CCI land cover maps based on the cross-walking table (Li et al., 2018). For the LUH2 dataset, we reclassified land use by summing forested primary land (primf) and potentially forested secondary land (secdf) to create a single "forest" category, and by summing all crop types (c3ann, c3per, c3nfx, c4ann, and c4per) to form the "cropland" category. To define natural grass and shrub, we first combined non-forested primary land (primn) and potentially non-forested secondary land (secdn) into a unified grass + shrub type. We then allocated this combined area back into separate grass and shrub categories based on their proportional distribution. For historical years before 1992, the proportional distribution was set the same as ESA CCI land cover in 1992, and for years in

1992-2020, the proportional distribution was set according to the corresponding year of ESA CCI land cover. The area fraction changes between two consecutive years from LUH2 were used to extrapolate the land use fraction in each year

before 1992. Therefore, the harmonized land use maps adopted the inter-annual variability from LUH2-GCB2020 before 1992 while the absolute area fractions were based on the ESA CCI maps. Comparisons of LUH2-GCB2020, ESA CCI maps and the harmonized maps in this study are shown in Fig. S3. Among the five land use types, four of them (forest, shrub, natural grass and cropland) were used as input to the machine learning models.

**Table 1: Explanatory variables used in the machine learning models.**

| Variables | Descriptions | Datasets | References |
|---|---|---|---|
| Tmax | Daily maximum temperature (°C) | CRU JRA v2.2 | (Harris et al., 2014; Harris et al., 2020; Kobayashi et al., 2015) |
| Tmin | Daily minimum temperature (°C) | | |
| Precip | Monthly total precipitation (mm/month) | | |
| Wind | Wind speed (m/s) | | |
| VPD | Vapor pressure deficit (hPa) | Calculated based on CRU JRA v2.2 | — |
| FWI | Fire weather index | | |
| Popd | Population density (individuals/km$^2$) | HYDE3.2 | (Klein Goldewijk et al., 2017) |
| LAI | Leaf area index in the previous months | TRENDY v11 GIMMS LAI4g | (Cao et al., 2023; Sitch et al., 2024) |
| Forest | Fraction of certain land use type in 0.5°×0.5° grid cells in each year | LUH2-GCB2020 CCI PFT map | (Chini et al., 2021; Li et al., 2018) |
| Shrub | | | |
| Natural Grass | | | |
| Cropland | | | |
| Δforest | Fraction difference of certain land use type in 0.5°×0.5° grid cells between the previous year and the current year | | |
| Δshrub | | | |
| Δnatural Grass | | | |
| Δcropland | | | |
| BAF | Burned area fraction in 0.5°×0.5° grid cells excluding burned area on cropland | FireCCI51 | (Lizundia-Loiola et al., 2020) |

We used the leaf area index (LAI) in the previous three months as a proxy of fuel status for the fire activity in the current month. Global monthly LAI maps in 0.5°×0.5° grid cells are resampled from GIMMS LAI4g, a satellite-based global LAI dataset available every half-month from 1982 to 2020 with a spatial resolution of 5 arc min (Cao et al., 2023). We further

generated LAI data from 1901 to 1981 after bias corrections using the multi-model average LAI from the simulations with dynamic climate, CO2 and land use change (S3 simulation) by eight DGVMs (EDv3, IBIS, ISAM, LPJ-GUESS, LPJmL, LPX-Bern, ORCHIDEE, and VISIT) in TRENDY v11 (Sitch et al., 2024). In the bias correction process, global monthly LAI difference (defined as LAI bias) maps between GIMMS LAI4g and multi-model averages from TRENDY v11 were firstly calculated from 1982 to 2020, and then machine learning models were utilized to predict the LAI biases using 15

variables (variables in Table 1 excluding LAI and BAF) from 1901 to 1981 after model training and testing with data covering 1982-2020 (80% for training and 20% for testing) in each region. Finally, the harmonized LAI during 1901-1981 is equal to the sum of multi-model average LAI from TRENDY v11 and the predicted LAI biases. LAI during 1982-2020 was directly derived from GIMMS LAI4g. The harmonized LAI maps therefore adopt the inter-annual and inter-monthly difference of TRENDY v11 to extrapolate the temporal coverage of GIMMS LAI4g to before 1982. Comparisons of LAI

from GIMMS LAI4g, TRENDY v11 and the harmonized data in this study are shown in Fig. S4-S6.

All datasets were aggregated into monthly data with a spatial resolution of 0.5°×0.5° for training and prediction of machine learning models.

## 2.2 Machine learning models

For each region (Fig. S1), we fed BAF as the dependent variable, and the 16 explanatory variables (Table 1) as independent

variables to build the machine learning models individually. To better capture the extreme fires, we firstly conducted a random forest classification model to distinguish grid cells (0.5°×0.5°) with no BAF, regular or extreme BAF. Extreme fire is usually defined by a percentile threshold of fire size, fire radiative power or fire spread rates in a region (Bowman et al., 2017; Castro Rego et al., 2021; Cunningham et al., 2024). Here, we defined grid cells with extreme BAF as grid cells with BAF exceeding the 90[th] percentile of all grid cells with fires in each region through the entire period (2003-2020). Other grid

cells with BAF greater than 0 were thus treated as grid cells with regular BAF. To balance sample sizes across BAF types, we applied a weighting method in machine learning classification models. Let the sample counts for no BAF, regular BAF, and extreme BAF be $n_1$, $n_2$, and $n_3$, respectively. We computed their least common multiple, M, and assigned weights of $M/n_1$, $M/n_2$, and $M/n_3$ to each BAF type. After classification, then we perform machine learning regressions separately for grid cells with regular or extreme BAF, and grid cells for each category (regular and extreme) are fed into separate

regression models to estimate the specific BAF value (continuous values).

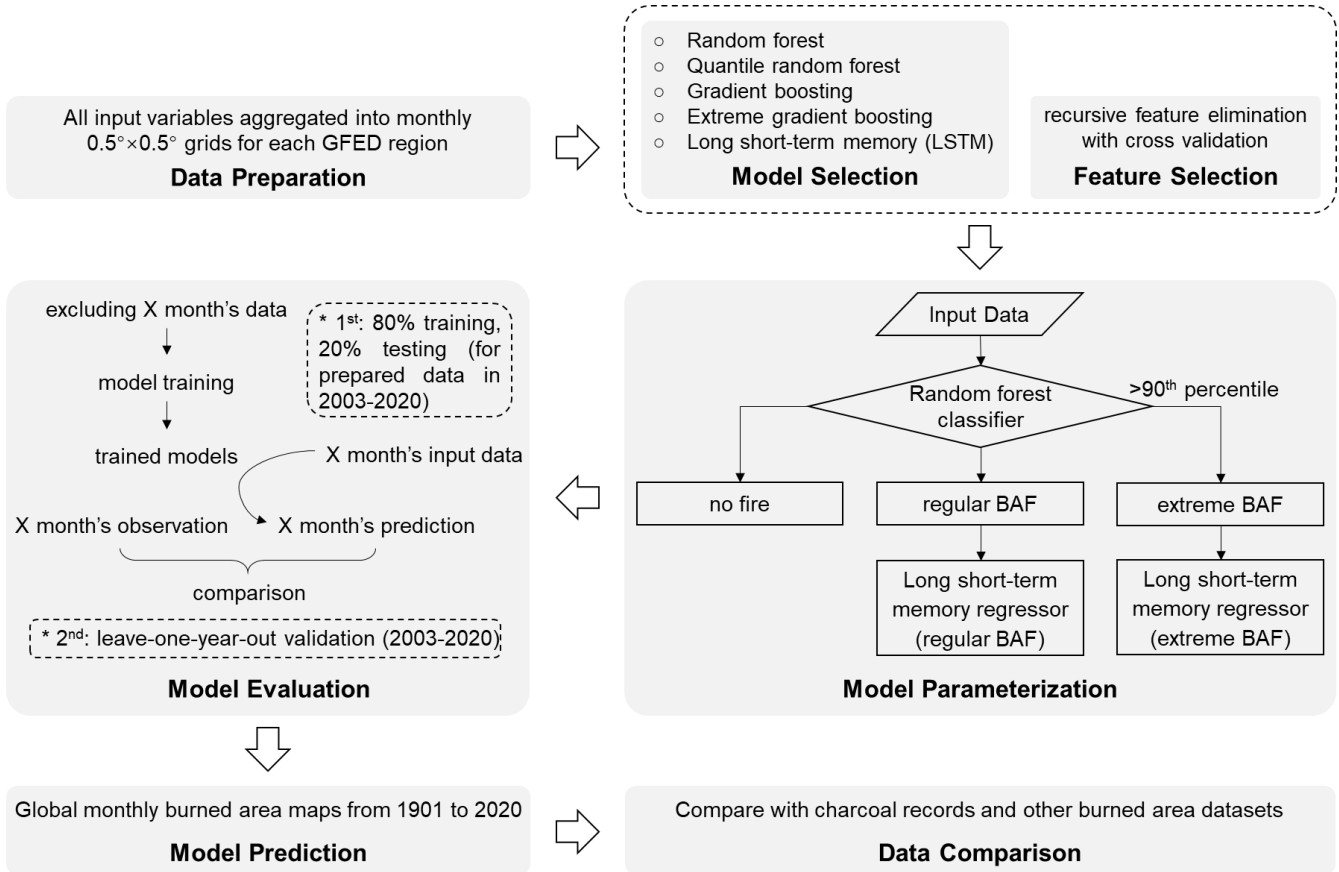

**Figure 1: Workflow of this study.**

For the regression model selection, we tested commonly used machine learning models including random forest (Tin Kam, 1995), quantile random forest (Meinshausen, 2006), gradient boosting (Friedman, 2001) and extreme gradient boosting (Chen and Guestrin, 2016), and a deep learning architecture called long short-term memory networks (LSTMs) (Hochreiter and Schmidhuber, 1997) in NHAF and BOAS. We chose NHAF as the testing region because its annual total burned area dominates global annual total burned area, and our preliminary tests severely underestimated the annual total burned area in NHAF, thus we aimed to improve model performance in NHAF by testing different machine learning models. In this test, we took only one year data (2010) and split it into the training set (80%) and the testing set (20%). In addition, we selected BOAS as another testing region because this region experiences regular fires but has different climate and vegetation conditions from NHAF. In this test, we took only one year data (2010) and split it into the training set (80%) and the testing set (20%). It turned out that LSTMs have the best performance (Fig. S7, S8, S10h, S11h) for regression with a memory window of three months. LSTMs consist of three gated memory cells (input gate, forget gate, output gate) that require the time series of input data can integrate information over long time series (Hochreiter and Schmidhuber, 1997), exhibiting

good performance on extreme events (e.g., precipitation, floods) (De Sousa Araújo et al., 2022; Nearing et al., 2024). All machine learning models were built using the 'scikit-learn' (Pedregosa et al., 2011) and the 'pytorch' (Paszke et al., 2019) packages in Python.

In addition to the 16 explanatory variables in Table 1, we conducted sensitivity tests by incorporating lightning (Kaplan and Lau, 2021) and terrain information (Danielson and Gesch, 2011) in each region (Supplementary Text3) to assess whether these variables can help improve model performance. We also tested other variables in NHAF (e.g., gross domestic product, human development index, livestock density, road density, tree cover, forest aboveground biomass) (Fig. S9), but they were excluded either using recursive feature elimination cross validation (i.e., negligible contributions to the model) or due to the

limited time span (i.e., not covering the entire $20^{th}$ century and difficult to extrapolate). The recursive feature elimination cross validation was applied to prevent model performance degradation if irrelevant features were added (Guyon and Elisseeff, 2003). Moreover, reducing the feature set could enhance model interpretability and conserve computational resources (Lundberg et al., 2020).

      For the model parameterizations, the timestep length was set to three consecutive months' (the previous two months and the

current month) in LSTMs to predict the current month's regular or extreme BAF. We randomly split the data over the period of 2003-2020 into five folds, using one fold (20%) as the testing set and the remaining four folds as the training set (80%). This process was looped for each of the five folds. We then used the training set to train the models and the testing set to evaluate model performance and optimized model parameters according to the principle of minimum Gini impurity for classification and minimum mean square error loss for regressions. We optimized model hyperparameters using a grid search

with five-fold cross-validation. For the Random Forest classifiers, we tuned "max_depth" and "n_estimators"; for the LSTM regressors, we tuned "hidden_sizes", "learning_rate", and "epochs". All combinations of these parameter values were used to retrain the models, and performance was evaluated on each held-out fold using coefficient of determination, slope, and rooted mean squared error. The combination yielding the best average metrics across folds was selected as optimal.

      After determining the optimal model parameters, we conducted the model evaluation using a leave-one-year-out method in

addition to the 5-fold evaluation method in the model parameterization process. Specifically for the period of 2003-2020 in each region, we excluded one year's data and used data from the remaining years to train the models, and data from this year were then used to compare with the models' predictions for this year. This procedure was repeated for each year during 2003-2020. The SHapley Additive exPlanations (SHAP) value, representing explainable contributions by features and their interactions in the machine learning models, was calculated with the 'shap' package (Lundberg et al., 2020) using tree

explainer and deep explainer for the random forest classification models and LSTMs, respectively, in Python.

      The machine learning models with optimal parameters from the 5-fold evaluation process were finally used to predict global monthly BAF maps from 1901 to 2020. For the time series of annual total burned area, we conducted breakpoint detections and linear regressions for each segment. The number of breakpoints were identified with Bayesian Optimization function in the 'GPyOpt' package (Javier Gonzalez, 2016), and linear regressions for each segment were conducted using the

PiecewiseLinFit function in the 'pwlf' package (Jekel and Venter, 2019) in Python.

## 2.3 Other fire datasets used for comparison

We used two databases of charcoal records, Global Charcoal Database v4 (Blarquez, 2020) and the Reading Palaeofire Database (Harrison et al., 2022) (D1 and D2 in Table 2), to evaluate the prediction accuracy of fire occurrence from 1901 to 2020. Fire occurrence in a certain grid cell refers to its BAF larger than 0, and we defined the prediction accuracy (%) for each site as the number of charcoal records that match our predicted fire occurrence divided by the total number of charcoal records multiplying 100%. Note that the charcoal age reported in both databases is associated with uncertainties, but only the Reading Palaeofire Database provides age uncertainties for some records. We thus calculated the average uncertainty across records with reported age uncertainties from 1901 to 2020 in the Reading Palaeofire Database and assigned this average uncertainty (3 years after rounding) to those records without reported uncertainties in both databases. For a given charcoal record, if there is a predicted fire occurrence in the same grid cell within the time span of the uncertainty age range, it is considered as a correct prediction.

In addition to the charcoal records databases, we compared our predicted burned area with burned area datasets that cover more than 10 years in different countries or regions (Table 2). Fire history reconstruction (D3 in Table 2) by Mouillot and Field (2005) is a global annual gridded (1°×1°) burned area dataset for the 20$^{th}$ century produced based on the regional burned area statistics. The datasets by the State Government of Australia (D11-D14 in Table 2) include wildfires and prescribed burns, mainly consisting of bureau statistics and missing data before the satellite-era and satellite-based data in the satellite-era. The remaining polygon and raster datasets (burned area products across the globe and in Canada, USA, Brazil, Chile and Europe) listed in Table 2 are all satellite-based. We converted the polygon data to the raster data at a spatial resolution of 30 m and resampled all raster data to 0.5°×0.5°. Then we calculated the time series of annual and monthly total burned area in each region or globally and also derived the spatial pattern averaged over all years of the reported period for each dataset.

## 3 Results

### 3.1 Model performance

In the optimization of model parameters (Sect. 2.2), the input data was randomly split into 80% for training and 20% for testing. Based this test subset, the overall accuracy of our random forest classification models ranges from 87.8% to 97.7% in the 14 regions, and the ranges of AUC (Area Under receiver operating characteristic Curves AUC ranging in 0-1 and a larger AUC value indicating better model performance) are 0.885-0.972 and 0.917-0.989 for grid cells with regular BAF and extreme BAF, respectively (Table S1). Based on the testing samples, the slopes of linear regression between predicted and observed regular BAF across all grid cells in each region range from 0.42 to 0.96, and the coefficients of determination ($R^2$) are between 0.60 and 0.95 (Fig. S10). The slopes for the extreme BAF are in the range of 0.43-0.96, and $R^2$ is between 0.58 and 1 (Fig. S11).

**Table 2: Charcoal and burned area datasets used for comparison.**

| No. | Datasets | Spatial Resolution | Temporal Resolution | Source |
|-----|----------|--------------------|---------------------|--------|
| D1 | Global Charcoal Database | Global, site | B.P. 21000- years | (Blarquez, 2020) |
| D2 | Reading Palaeofire Database | Global, site | B.P. 56000- years | (Harrison et al., 2022) |
| D3 | Fire History Reconstruction for the 20th century | Global, 1°×1° | 1900-1999, annual | (Mouillot and Field, 2005) |
| D4 | FireCCILT11 | Global, 0.05°×0.05° | 1982-2018, monthly | (Otón et al., 2021) |
| D5 | Global Annual Burned Area Maps (GABAM) | Global, 30m | 1985-2020, annual | (Long et al., 2019) |
| D6 | Monitoring Trends in Burn Severity | USA, polygon | 1984-2022, daily | (Eidenshink et al., 2007) |
| D7 | MapBiomas Fire Collection | Brazil, 30m | 1985-2020, annual | https://www.mapbiomas.org |
| D8 | Canadian National Burned Area Composite | Canada, polygon | 1986-2022, daily | https://cwfis.cfs.nrcan.gc.ca/datamart |
| D9 | European Forest Fire Information System | Europe, polygon | 1980-2020, daily | https://forest-fire.emergency.copernicus.eu |
| D10 | Occurrence of Wildfires by Time Range | Chile, polygon | 1985-2020, annual | Chilean National Forest Service (CONAF) |
| D11 | NPWS Wildfires and Prescribed Burns | New South Wales, polygon | 1935-present, annual | https://datasets.seed.nsw.gov.au/dataset/ |
| D12 | Fire history – Queensland Parks and Wildlife Service | Queensland, polygon | 1937-present, annual | https://qldspatial.information.qld.gov.au/catalogue/custom |
| D13 | Bushfires and Prescribed Burns History of South Australia | South Australia, polygon | 1950-present, annual | https://data.sa.gov.au/data/dataset/fire-history |
| D14 | DPCA Fire History of Western Australia | Western Australia, polygon | 1926-present, annual | https://catalogue.data.wa.gov.au/dataset/dbca-fire-history |

In the model evaluation based on the leave-one-year-out method, the multi-year (2003-2020) mean burned area fraction (BAF) between observation (FireCCI51) and our prediction is spatially consistent in general. There is a strong spatial correlation between observations and prediction at the global scale with $R^2$ of 0.97 and a linear slope of 0.97. Among all regions, $R^2$ ranges from 0.69 to 0.98, and slopes range from 0.70 to 1.28 (Fig. 3a-3o), which indicates that the trained models can well reproduce the spatial patterns of burned area. Still, some regions show mismatches, especially in the tropics (Fig. 2a-b). Our predictions tend to overestimate BAF in the southeastern regions in South America (Fig. 2a, 3f) and in the southern part of Africa but underestimate BAF in North Africa (Fig. 2a, 3i), though relative difference between predictions

and observations is small in these regions. BAF in boreal North America and boreal Asia is also partially underestimated by our predictions (Fig. 2a, 3b, 3k). Large relative differences exist in the boreal regions compared with observations (Fig. 2b) due to the smaller absolute burned area than the tropical regions. Additionally, the relative differences within 40-60 °S fluctuate due to a small number of land grid cells with fire occurrence (Fig. 2b).

The interannual variabilities of global total burned area during 2003-2020 between predictions and observations are in a good agreement at the global scale and within each region (Fig. S12). $R^2$ from the temporal regressions of global total burned area between predictions and observations is 0.88, and it ranges from 0.43 to 0.99 across regions. The linear slope is 0.79 for the global total burned area, and its range is 0.54-1.24 across regions. The model can also well capture the seasonality of burned area in each region (Fig. S13, S14).

## 3.2 Variable importance for predicting burned area

We also analyzed the variable importance based on the SHAP values of predicting burned area. In terms of similarities in variable importance between the classification and regression models, fire weather index (FWI) and leaf area index (LAI) in the previous months are among the most important variables in most regions, while the land use change fraction between the previous year and the current year (Δfraction) has low importance in all regions.

In the classification models built for distinguishing grid cells with no fire, regular BAF or extreme BAF (Fig. 4a), FWI and LAI, indicating climatic conditions and vegetation status respectively, are the two most important predictors across the globe. Vapor pressure deficit (VPD) and daily maximum temperature (Tmax) are the most important climatic factors in boreal regions (BONA, BOAS). The area fractions of natural grass and cropland rank in the top five in MIDE, NHAF, CEAS and EQAS. The contribution of precipitation (Precip) is high in the tropical regions (NHSA, NHAF, SHAF and EQAS). Wind speed (Wind) is consistently not essential in all regions. The importance of population density (Popd) remains low in most regions except in EURO and CEAS.

In the regression models predicting regular BAF and extreme BAF (Fig. 4b, 4c), the contribution of FWI is larger in predicting extreme BAF than regular BAF, while Tmax, VPD and LAI in the previous months are more crucial in predicting regular BAF than extreme BAF. Popd consistently shows low importance on predicting both regular BAF and extreme BAF. The cropland area fraction has the largest contribution in NHAF and CEAS for predicting regular BAF, and forest area fraction is the most important variable for predicting extreme BAF in BONA and SEAS. Additionally, the shrub area fraction ranks in top 5 for predicting extreme BAF in NHAF, SHSA and AUST.

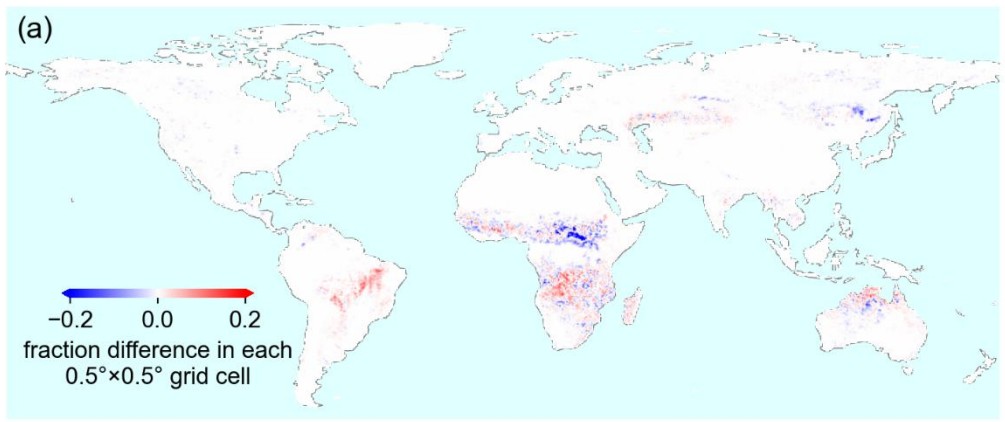

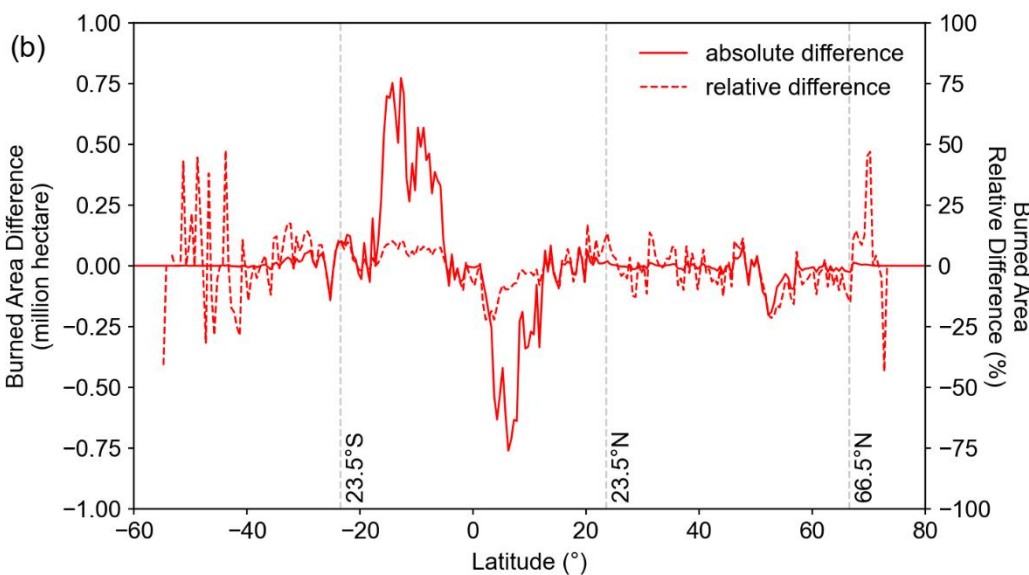

**Figure 2: Multi-year (2003-2020) averaged burned area difference between our predictions by the leave-one-year-out method and FireCCI51 observations (predictions minus observations). (a) Map of burned area fraction difference in each 0.5°×0.5° grid cell. Burned area fraction difference is the ratio of burned area difference to total grid area within each 0.5°×0.5° cell, making it unitless and bounded between 0 and 1. (b) Latitudal sum of burned area difference using the burned area fraction difference map from (a) multiplied by the area of each 0.5°×0.5° grid cell. Both absolute (solid line) and relative (dashed line) differences are shown.**

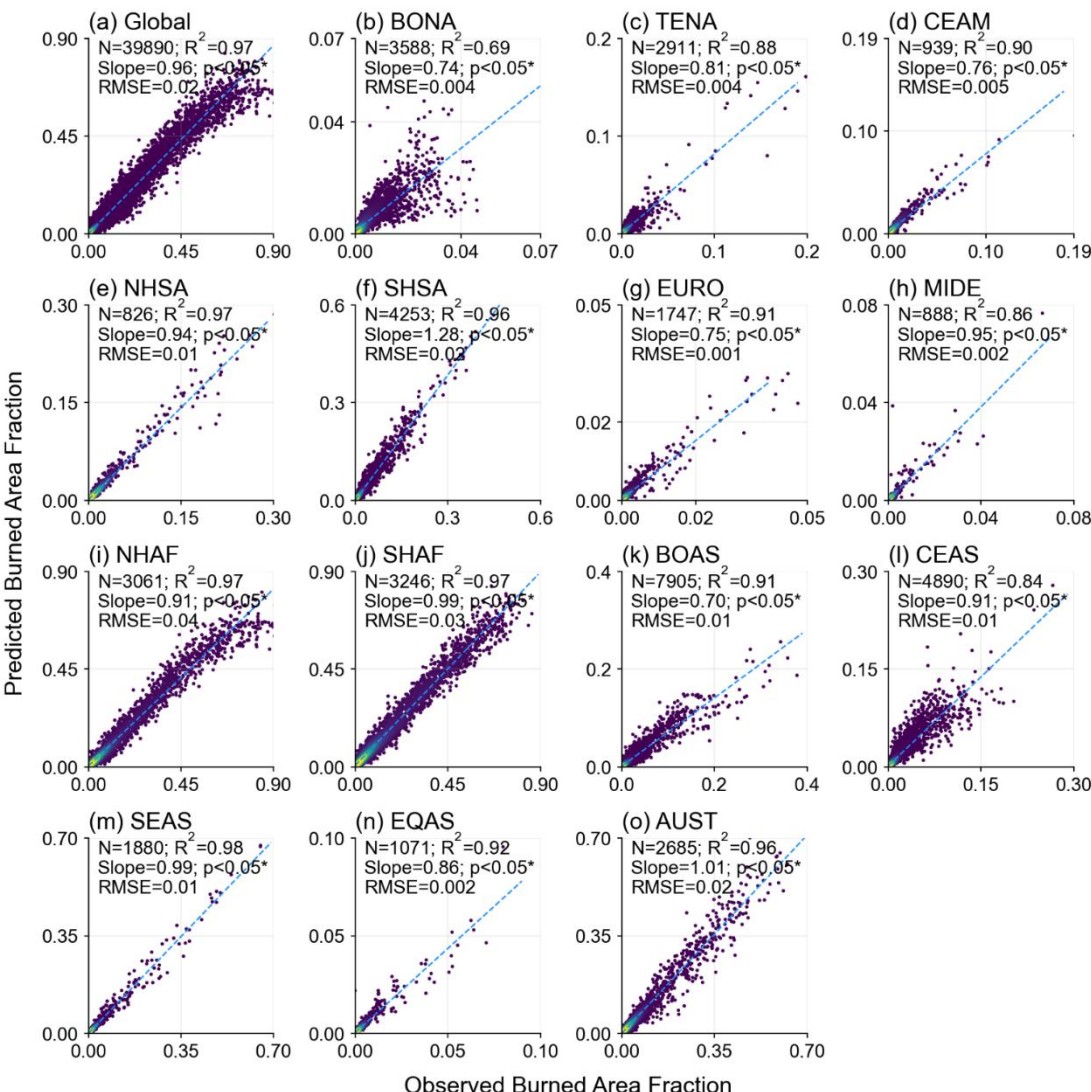

**Figure 3: Scatter plots of multi-year (2003-2020) averaged burned area fraction (BAF) in each 0.5°×0.5° grid cell from predictions by the leave-one-year-out method and FireCCI51 observations for each region (a-o). Dots represent grid cells with BAF>0 averaged over 2003-2020. N, R², slope, p and RMSE respectively represent number of grid cells with multi-year averaged BAF>0, coefficient of determination, linear slope, p-value for linear correlation and rooted mean squared error between BAF from our predictions and observations. Burned area fraction is the ratio of burned area to total grid area within each 0.5°×0.5° cell, making it unitless and bounded between 0 and 1.**

300

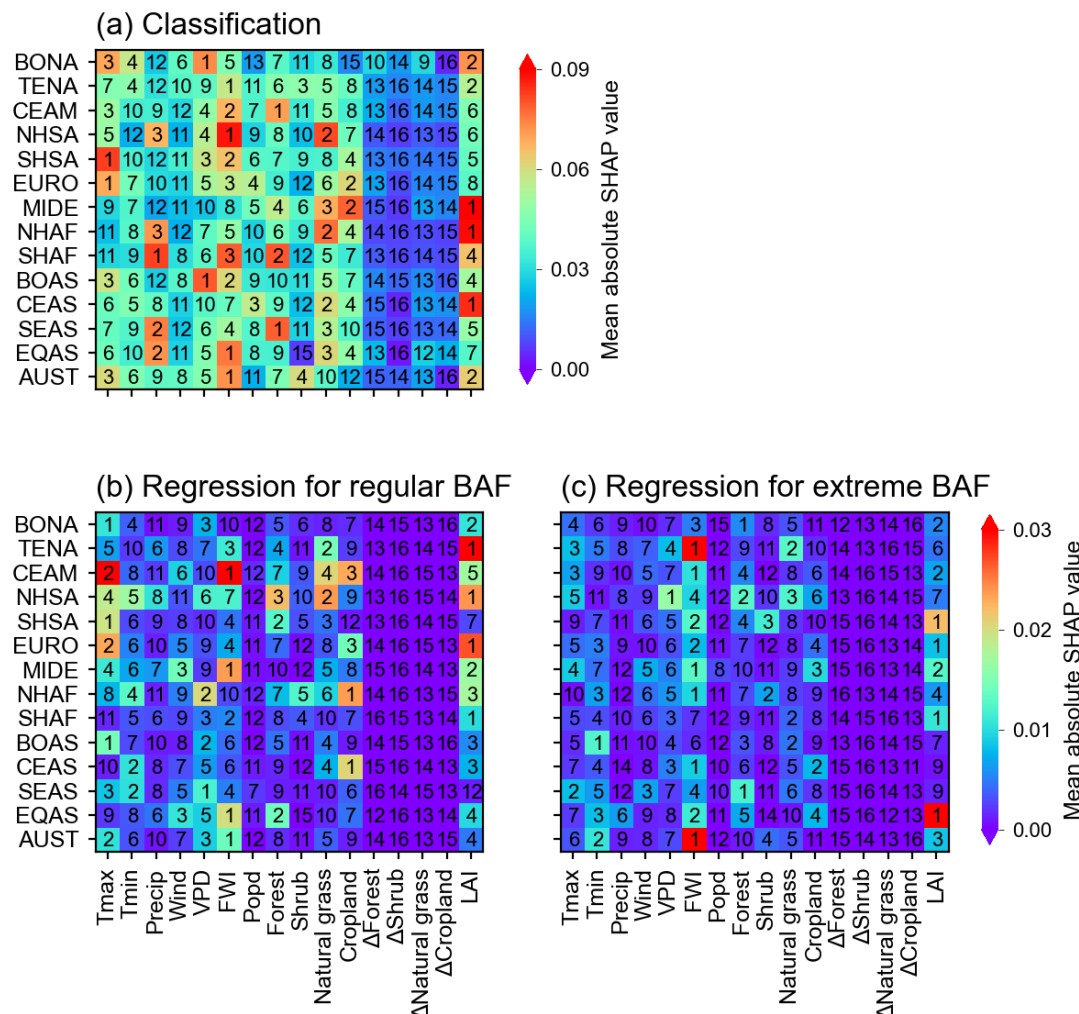

**Figure 4: Mean absolute SHAP value (a,c,e) and the ranking (b,d,f) of all input variables (Table 1) using the random forest classification models (a,b) and the LSTMs regression models for regular (c,d) and extreme (e,f) BAF, respectively, in each region. Higher ranking (i.e., smaller rank number, redder) represents relative higher mean absolute SHAP value in the corresponding GFED region. Numbers denoted in grids are the ranking of variables, and higher ranking represents relative higher mean absolute SHAP value in the corresponding GFED region.**

## 3.3 Predicted burned area

The predicted annual global total burned area from 1901 to 2020 ranges from 3.46 to 4.58 million $km^2$ per year (M $km^2$ $yr^{-1}$), with regular and extreme burned area accounting for 1.36-1.74 M $km^2$ $yr^{-1}$ and 2.00-3.03 M $km^2$ $yr^{-1}$ respectively (Fig. S15a, S16a). By comparison, the global total burned area from FireCCI51 is in the range of 3.43-4.58 M $km^2$ $yr^{-1}$ over 2003-2020 (1.43-1.71 and 1.99-2.98 M $km^2$ $yr^{-1}$ for regular and extreme fires) (Fig. S15a, S16a). Breakpoint detection shows three

segments with two breakpoints around 1978 and 2008 globally (Fig. 5a). Global total burned area was decreasing from 1901 to 1978 (slope = -0.009 M km$^2$ yr$^{-2}$), increasing from 1978 to 2008 (slope = 0.020 M km$^2$ yr$^{-2}$) and then decreasing again from 2008 to 2020 (slope = -0.049 M km$^2$ yr$^{-2}$) (Fig. 5a), whereas the above trends are mainly contributed by extreme burned area, with global total extreme burned area decreases from 1901 to 1978 (slope = -0.007 M km$^2$ yr$^{-2}$), increases from 1978 to 2008 (slope = 0.019 M km$^2$ yr$^{-2}$) and decreases again from 2008 to 2020 (slope = -0.047 M km$^2$ yr$^{-2}$) (Fig. S16a, S17a). Northern Hemisphere Africa (NHAF) and Southern Hemisphere Africa (SHAF) are the top two regions with the largest annual total burned area. The annual total burned area of NHAF is in the range of 0.97-1.90 M km$^2$ yr$^{-1}$, and it was increasing from 1901 to 1922 (slope = 0.005 M km$^2$ yr$^{-2}$), declining from 1922 to 1957 (slope = -0.008 M km$^2$ yr$^{-2}$) and then declining more slowly from 1957 to 2020 (slope = -0.004 M km$^2$ yr$^{-2}$) (Fig. 4i), dominated by extreme burned area (Fig. S16i, S17i). The annual total burned area of SHAF is in the similar range (1.15-1.87 M km$^2$ yr$^{-1}$) as NHAF, and it also shows a similar decreasing trend from 1901 to 1979 (-0.004 M km$^2$ yr$^{-2}$), but it turned into increasing from 1979 to 2011 (slope = 0.011 M km$^2$ yr$^{-2}$) and then decreasing from 2011 to 2020 (slope = -0.019 M km$^2$ yr$^{-2}$) (Fig. 5j), dominated by regular burned area (Fig. S16j, S17j). Therefore, the global burned area trends (Fig. 5a) are predominantly controlled by the trends in SHAF (Fig. 5j).

The total burned area in other tropical regions such as southern hemisphere South America (SHSA) and equatorial Asia (EQAS) is lower than that in Africa. The annual total burned area in SHSA is dominated by extreme burned area (Fig. S17f) and varies from 0.09 to 0.66 M km$^2$ yr$^{-1}$, and it kept increasing from 1901 to 1962 (slope = 0.001 M km$^2$ yr$^{-2}$), decreasing with a slope of -0.002 M km$^2$ yr$^{-2}$ from 1962 to 1974 and then increasing from 1974 to 2020 (0.004 M km$^2$ yr$^{-2}$) (Fig. 5f). The annual total burned area in EQAS, dominated by extreme burned area (Fig. S17n), ranges from 0.002 to 0.018 M km$^2$ yr$^{-1}$, but no significant trends were detected. In the boreal regions, the trends of annual total burned area are different between boreal North America (BONA) and boreal Asia (BOAS). The annual total burned area was increasing at 0.0003 M km$^2$ yr$^{-2}$ from 1901 to 1929 but decreasing at -0.0002 M km$^2$ yr$^{-2}$ from 1927 to 2017 in BONA (Fig. 5b). By contrast, it was decreasing at -0.0006 M km$^2$ yr$^{-2}$ from 1901 to 1941 but increasing at 0.0009 M km$^2$ yr$^{-2}$ from 1941 to 1997 in BOAS (Fig. 5k).

### 3.4 Comparison with charcoal records and other burned area datasets

Two charcoal databases, Global Charcoal Database and Reading Palaeofire Database, were applied to calculate the overall accuracy (%) of the predicted fire occurrence. The overall accuracy of two databases is 41.4%±23.8% and 33.0±15.3% (average and standard deviation of accuracy values among 1901-2020), respectively (Fig. 6c). Spatially, the number of sites in the Reading Palaeofire Database (Fig. 6b) is larger than that in the Global Charcoal Database (Fig. 6a). Sites with high accuracy (> 80%) mainly locate in boreal North America, southern Europe, South America, Africa and southeast Australia (Fig. 6a, 6b). However, sites in northern Europe and Asia have relatively lower accuracy (<50%). In addition, the accuracy across the globe using the Global Charcoal Database exhibits a significant increasing trend from 1901 to 2020, indicating better model performance in the recent period (Fig. 6c).

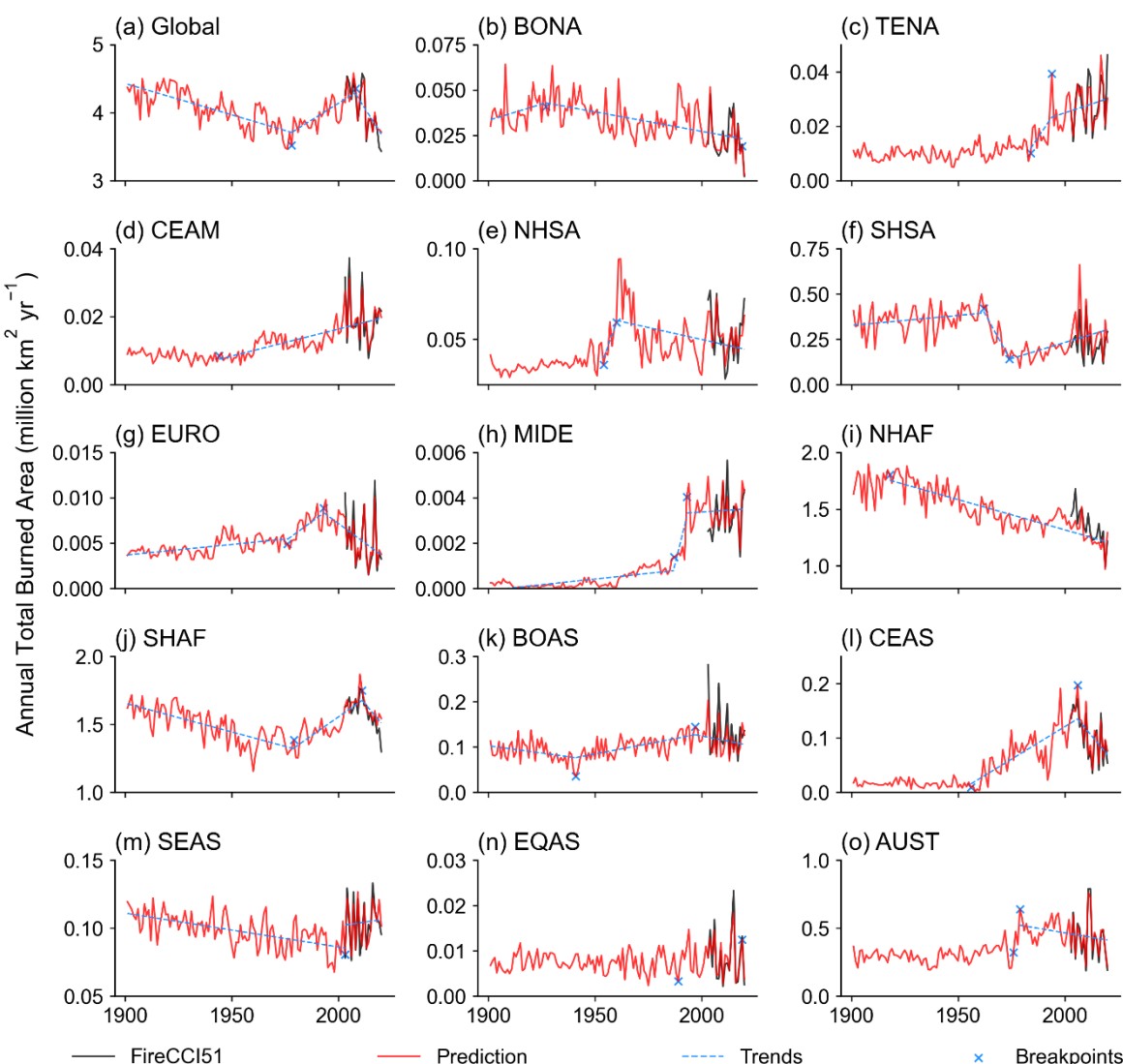

**Figure 5: Time series of annual total burned area across the globe (a) and in each region (b~o) from FireCCI51 (black lines, 2003~2020) and predictions (red lines, 1901~2020). The breakpoints and significant slopes (p-value < 0.05) in blue were also shown (Sect. 2.2).**

We further compared our predicted burned area with independent burned area datasets at the global and regional scale (Table 2). Significant trends of annual total burned area in different regions from various datasets are summarized in Table S2.

At the global scale, we first compared our predictions with FireCCILT11, a satellite-based burned area dataset available monthly from 1982 to 2018 (1994 was missing) (Otón et al., 2021). Spatially, our predicted multi-year average BAF is lower than FireCCILT11 in regions such as CEAS, SEAS, Africa and the southern part of Australia (Fig. 7a). Consequently, the global and regional annual total burned area from our predictions is lower than that from FireCCILT11 (Fig. 8a-8o). The

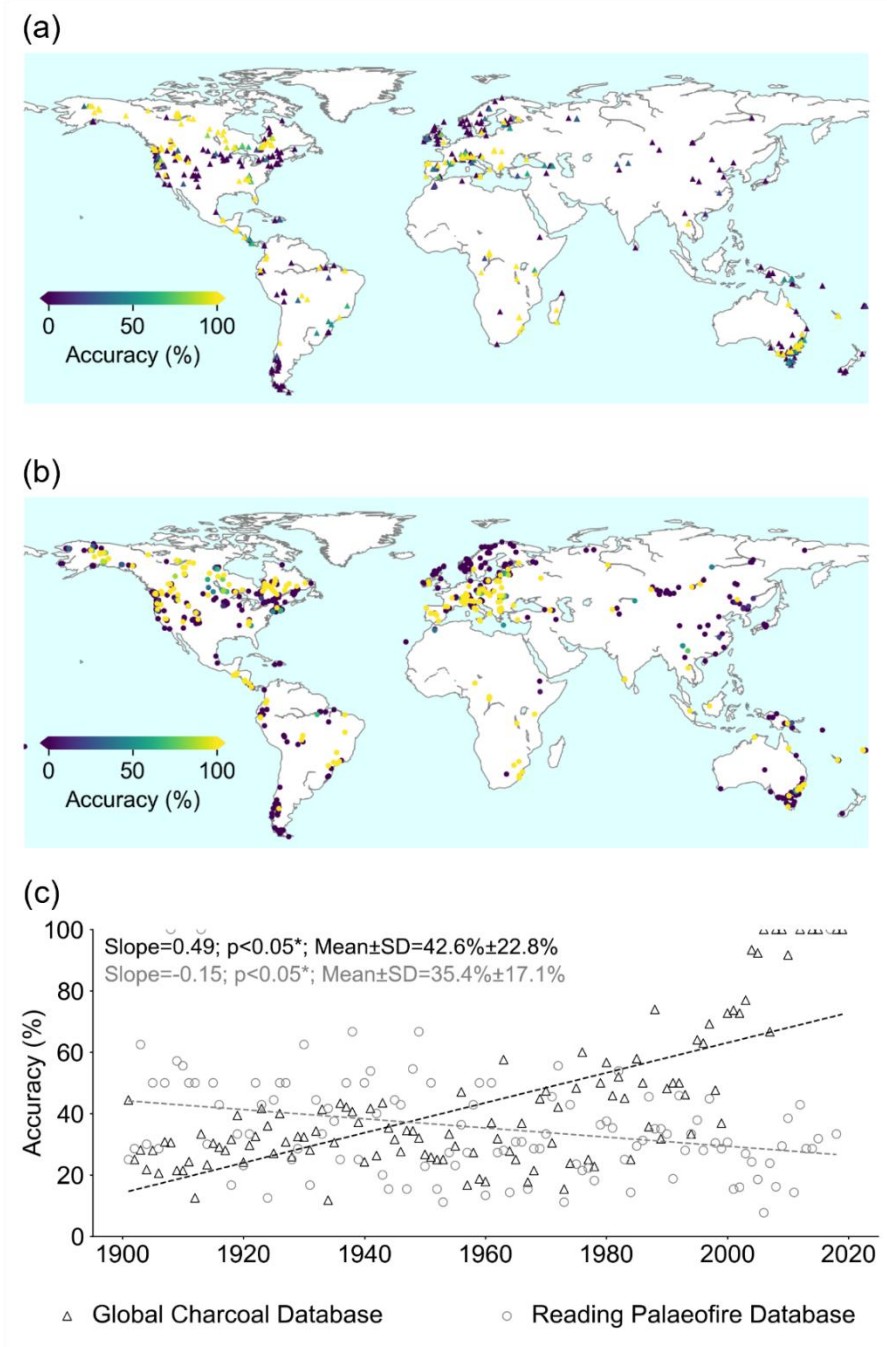

**Figure 6: Fire occurrence comparison between two charcoal record databases and our prediction from 1901 to 2020. (a) Site accuracy map using Global Charcoal Database. The site accuracy (%) is equal to the number of records with predicted burned area dividing the number of all records multiplying by 100%. (b) The same as (a) but using Reading Palaeofire Database instead. (c) Accuracy time series using Global Charcoal Database and Reading Palaeofire Database respectively. Note that only records with record year ± record age uncertainty overlapping with 1901-2020 are taken into consideration.**

global annual total burned area from our predictions ranges from 3.61 to 4.58 M km$^2$ yr$^{-1}$ during 1982-2018, compared to 4.09-5.18 M km$^2$ yr$^{-1}$ from FireCCILT11 over the same period. The significant linear trends were detected in the time series of annual total burned area from FireCCILT11 at the global scale and in CEAM, SHSA, MIDE, NHAF, SHAF, CEAS, and the trends are generally comparable to our predictions (Table. S2). It should be noted, however, significant orbit-drift artifacts may cause biases in the FireCCILT11 product over numerous large spatial patches almost on every continent except Antarctica (Giglio and Roy, 2022, 2024), and thus these comparisons should be interpreted with caution, especially in the tropics.

Next, we also compared our predictions with Global Annual Burned Area Maps (GABAM) (Long et al., 2019), a satellite-based burned area dataset available almost annually from 1985 to 2020 except in 1986, 1988, 1990, 1991, 1993, 1994, 1997 and 1999. Spatially our predicted multi-year average BAF is higher than GABAM mainly in the tropical regions (most Africa, Amazon and northern part of Australia) (Fig. 7b). This is because GABAM was produced from the Landsat imagery, which has a lot of missing data in the tropics, and thus underestimated burned area in these regions (Long et al., 2019; Pessôa et al., 2020). During 1985-2020, the global annual burned area from our predictions ranges in 3.61-4.58 M km$^2$ yr$^{-1}$, compared to 0.77-4.90 M km$^2$ yr$^{-1}$ from GABAM. Regionally, annual total burned area from GABAM is consistent with our predictions in BONA, TENA, CEAM and EQAS (Fig. 8b-d, 8n), but it is much higher in EURO and MIDE (Fig. 8g, 8h). As a global dataset, neither evidence of active fires or region-specific algorithms was taken into consideration in GABAM, which could also introduce uncertainty of burned area detection (Long et al., 2019).

We further compared our predictions with the fire reconstruction dataset by Mouillot and Field (2005) based on regional statistics. The multi-year average BAF from our predictions is lower than that from Mouillot and Field (2005) in the southeastern USA, SHSA, NHAF, India and Australia, but it is higher in SHAF and BOAS (Fig. 7c). The range of global annual total burned area during 1901-1999 is 3.46-4.51 M km$^2$ yr$^{-1}$ from our predictions, compared to 3.80-7.23 M km$^2$ yr$^{-1}$ from Mouillot and Field (2005) (Fig. 8a). At the global scale, annual total burned area increased respectively at 0.020 M km$^2$ yr$^{-2}$ (Fig. 4a) in 1978-2008 and 0.041 M km$^2$ yr$^{-2}$ in 1972-1997 from Mouillot and Field (2005) (Fig. 8a). Annual total burned area from our predictions and Mouillot and Field (2005) exhibit similar trends in some regions. For example, the decreasing trend of annual total burned area in BONA during 1929-2017 from our predictions (Fig. 5b) is consistent with the trend during 1920-1965 from Mouillot and Field (2005) (Fig. 5b, 8b). In SHSA, similar increasing trends were detected during 1974-2020 from our predictions and during 1973-1999 from Mouillot and Field (2005) (Fig. 5f, 8f).

Comparing with the regional burned area datasets in Canada, USA (continental USA and Alaska), Brazil, Chile and Europe, the predictions generally reproduce the inter-annual variability of the total observed burned area (Fig. 9a-9f). The slopes of multi-year average burned area between our predictions and regional datasets range from 0.84 to 1.38, suggesting that they are in a good agreement spatially (Fig. 9k-9p). Especially in Brazil and Chile, our predictions are highly consistent with the burned area from MapBiomas (D7, Table 2) and CONAF (D10, Table 2) respectively based on the Landsat satellites, even in the period before 2000 when no burned area observation was used to train our models (Fig. 9d, 9e).

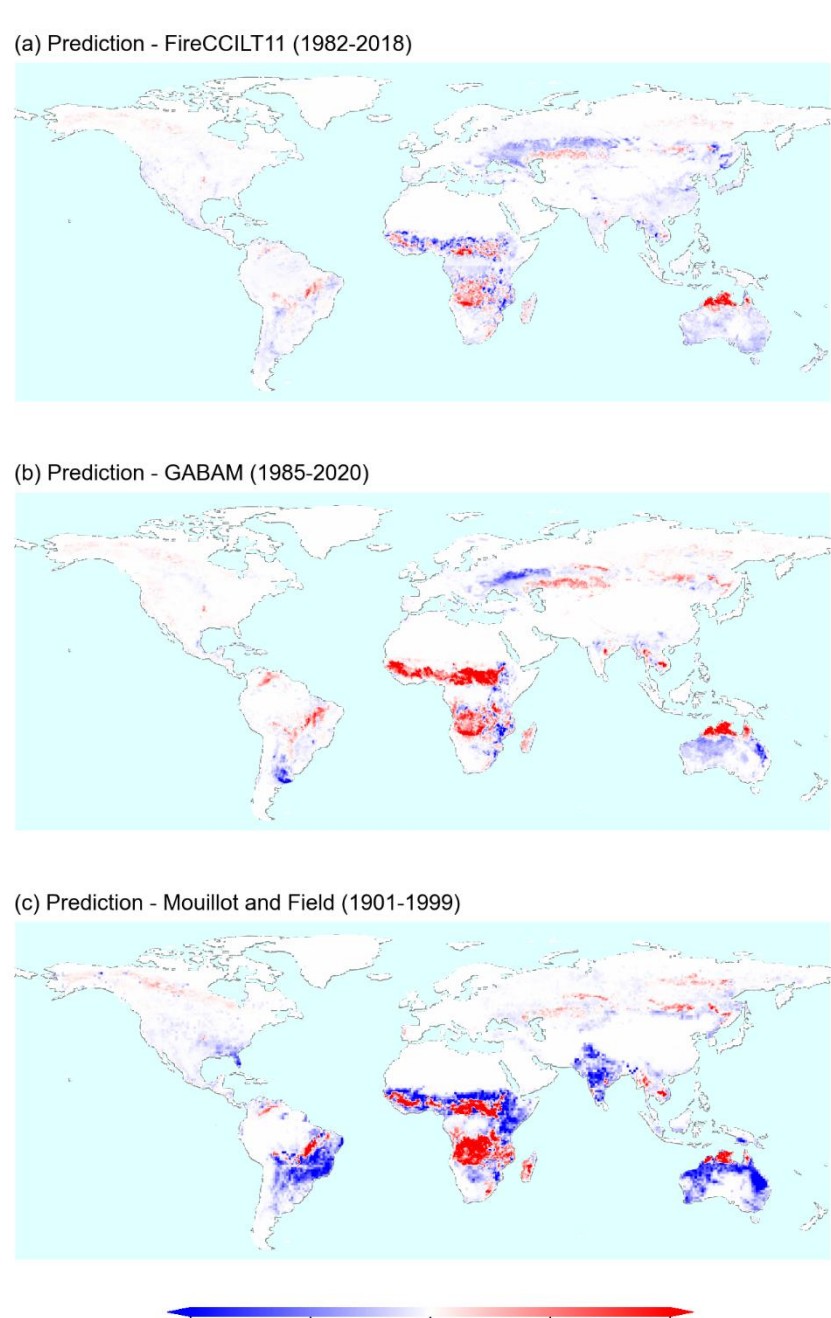

**Figure 7: Maps of burned area fraction difference between our predictions and other global burned area datasets. (a) Map of multi-year average (1982-2018) burned area fraction difference map between our predictions and FireCCILT11 (the former minus the latter). (b, c) Same as (a) but using Global Annual Burned Area Maps (GABAM) (1985-2020) and Mouillot and Field (2005) (1901-1999) instead respectively. Note that there are several years (1986, 1988, 1990, 1991, 1993, 1994, 1997 and 1999) without available data before 2000 in GABAM.**


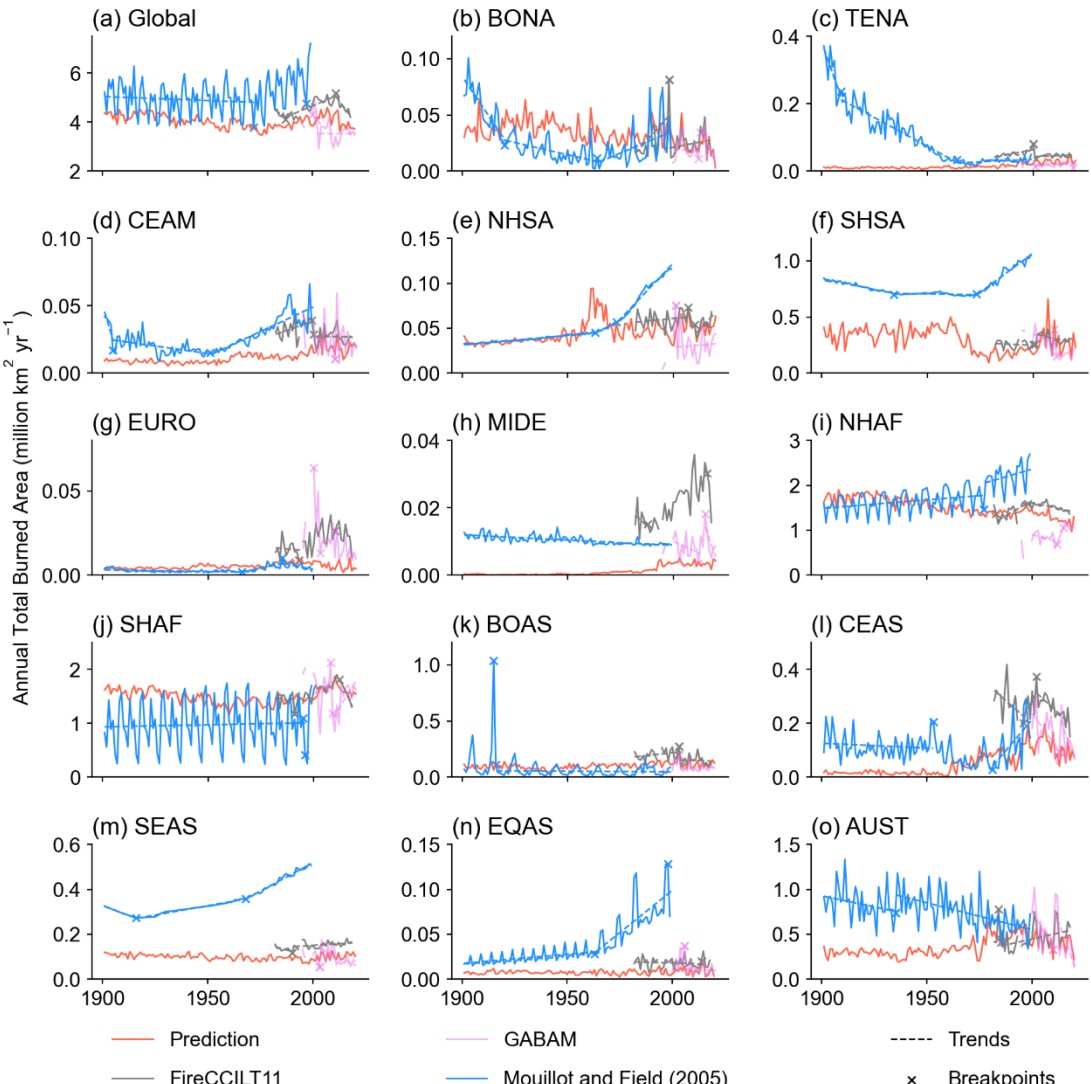

**Figure 8: Time series of annual total burned area across the globe (a) and in each region (b-o) from our predictions (red lines), Mouillot and Field (2005) (blue lines), FireCCILT11 (grey lines) and GABAM (purple lines). The breakpoints and significant slopes (p-value<0.05) were calculated by methods mentioned in Sect. 2.2. Note that there are several years (1986, 1988, 1990, 1991, 1993, 1994, 1997 and 1999) without available data before 2000 in GABAM, and thus breakpoint detection and linear slopes were applied after 2000 for this dataset.**

Our predicted burned area is higher than that from the regional datasets in USA and Europe before 2000 (Fig. 9b, 9f), probably because our model only included anthropogenic factors such as land use, land use change and population density, but some fire managements such as suppression in these regions were not explicitly included (Table 1). In Queensland and Western Australia (Fig. 9h, 9j), burned area from our predictions and FireCCI51 are also both much higher than that reported

burned area from the state governments (D11-D14, Table 2). Compared to the regional burned area datasets in Australia, our
predictions are higher in the northern part but lower in the southern part (Fig. 9u). Although the time span of the burned area datasets in Australia is long, the incomplete statistics and missing data before the satellite-era may also influence the reliability of these datasets.

## 3.5 Additional product versions

In addition to the historical reconstructed burned area dataset based on the FireCCI51 presented above, we also produced
two additional products of historical burned area with the same spatiotemporal resolution as the FireCCI51-based burned area reconstruction: 1) the GFED5-based data version, which is based on machine-learning models trained by the burned area from GFED5 which has much more fires than GFED4 (Chen et al., 2023) instead of FireCCI51, and 2) the FireCCI51-based data with burned area further calibrated using the relationship between statistic-based burned area (Mouillot and Field, 2005) and GDP (Bolt and Van Zanden, 2024) at the regional scale before 2000 (named as FireCCI51-GDP version).

The GFED5-based data version was produced and validated using the same methods as Sect. 2.1 and 2.2 but replacing the burned area of FireCCI51 with GFED5. The GFED5 burned area is based on the MCD64A1 burned area and adjusted by the Landsat or Sentinel-2 data, including more small fires in this product (Chen et al., 2023). The evaluation results of GFED5-based reconstruction are explicitly described in Supplementary Text 1 (Fig. S18-S20). Briefly, the annual global total burned area in the GFED5-based data version from 1901 to 2020 ranges from 5.42 to 7.35 M $km^2$ $yr^{-1}$, with regular and extreme
burned area accounting for 2.72-3.13 M $km^2$ $yr^{-1}$ and 2.54-4.27 M $km^2$ $yr^{-1}$, respectively (Fig. S21a, S22a, S23a), compared to the range of 3.46-4.58 M $km^2$ $yr^{-1}$ over 1901-2020 (1.36-1.74 and 2.00-3.03 M $km^2$ $yr^{-1}$ for regular and extreme fires) in the FireCCI51-based reconstruction (Fig. S15a, S16a). In most regions, despite the annual total burned area from GFED5-based reconstruction is generally higher than that from FireCCI51-based reconstruction caused by different data sources (e.g., more small fires from GFED5), the trends of annual total burned area from both reconstructions are generally consistent
across different regions (Fig. S21). However, one main difference between these two data versions is that the decreasing trend of global annual total burned area in the first half of the 20[th] century disappear in the GFED5-based reconstruction (Fig. S21a) because of the diminished decreasing trend in the SHAF (Fig. S21j).

To explicitly consider more anthropogenic effects (e.g., fire suppression, landscape fragmentation) in addition to the population density used in the original FireCCI51-based reconstruction, we also calibrated the reconstructed burned area
before 2000 at the regional scale using GDP as a proxy of anthropogenic effects and the statistic-based burned area from Mouillot and Field (2005) (see detailed methods in Supplementary Text 2 and Fig. S24). Temporally, the annual global total burned area in the FireCCI51-GDP data version before 2000 ranges from 4.77 to 6.44 M $km^2$ $yr^{-1}$, with regular and extreme burned area accounting for 1.68-2.31 M $km^2$ $yr^{-1}$ and 2.87-4.29 M $km^2$ $yr^{-1}$ respectively (blue lines in Fig. S21a, S22a, S23a). The area is much larger than the global total burned area from the original FireCCI51-based data version which ranges in
3.46-4.51 M $km^2$ $yr^{-1}$ before 2000 (1.41-1.74 and 2.00-2.95 M $km^2$ $yr^{-1}$ for regular and extreme fires) (Fig. S15a, S16a). The temporal trends of annual total burned area in the FireCCI51-GDP version is similar with the original FireCCI51-based

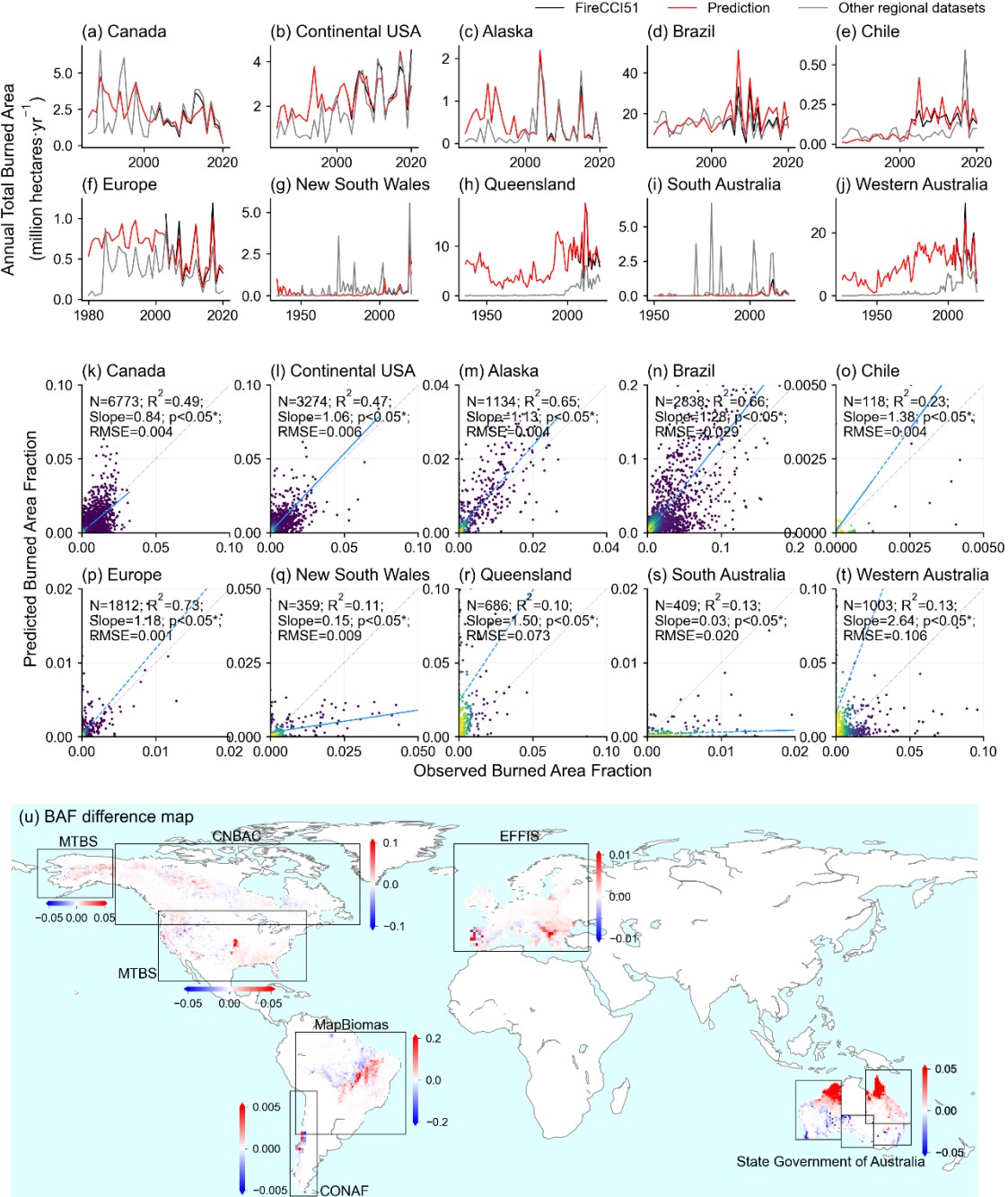

**Figure 9: Burned area comparison between our predictions and other regional burned area datasets. (a-j) Time series of annual total burned area from FireCCI51 (black lines), our predictions (red lines) and other regional datasets (grey lines). (k-t) Scatter plots of burned area fraction between our predictions and other regional datasets using multi-year averaged values (the full time span of observations). (u) Map of multi-year average burned area difference between our predictions and other regional datasets (the former minus the latter). The labels next to each rectangle correspond to the abbreviated names of datasets in Table 2. Note that scales of color bars are different across regions.**

version in most regions. However, the trends in TENA (Fig. S21c), NHAF (Fig. S21i) and AUST (Fig. S21o) are opposite between the FireCCI51-GDP version and the original FireCCI51-based version. It could be partly explained by the trends of annual total burned area from Mouillot and Field (2005) (Fig. 8i, 8o), which was used to build the relationship between regional burned area and GDP. The regional total burned area after calibration by GDP was applied proportionally to each grid cell based on the gridded burned area from the original FireCCI51-based version (Supplementary Text 2). As a result, the spatial patterns remain similar between these two data versions.

## 4 Uncertainty and discussion

Although our models show good performance in the evaluation (Fig. 2-3, S10-13), there are some uncertainties in the reconstructed historical burned area product associated with the input explanatory data, satellite-based burned area used for model training and the model selection. Due to the limited time span of the satellite-based data such as land use change and LAI, we harmonized the satellite-based datasets with other datasets before the satellite era. For example, large regional and global differences were found between the two land use datasets (LUH2-GCB2020 and CCI PFT maps) (Fig. S3). Different data sources, definitions of land use types and uncertainties in the cross-walking table for converting the 37 original ESA CCI land cover types to the main land cover classes may partly explain the differences between two land use datasets (Li et al., 2018). There are several alternative land cover and land use datasets with higher spatial resolution than ESA CCI and LUH2-GCB2020, such as forest maps (Hansen et al., 2013; Vancutsem et al., 2021) and global cropland extent maps (Potapov et al., 2022). However, due to the limited temporal coverage or land cover types, we used ESA CCI and LUH2-GCB2020 for their consistent and comprehensive land cover and land use types and the long temporal coverage. Differences also exist between the two LAI datasets (GIMMS LAI4g and LAI from TRENDY v11) (Fig. S5, S6) due to the different data sources. GIMMS LAI4g is a satellite-based product (Cao et al., 2023), while LAI from TRENDY v11 was simulated by DGVMs (Sitch et al., 2024). In addition to features selected in this study, other variables besides Table 2 were tested but eliminated (Fig. S9, Table S3-S5). For instance, lightning data is only available during 2010-2024, and thus it cannot be utilized to reconstruct burned area in the 20th century (Kaplan and Lau, 2021). Terrain resampling to 0.5°×0.5° grid cells inevitably diluted explicit information from a fine spatial resolution (Cary et al., 2006), thus posing minor effect across all regions except NHAF and SHAF. Gross domestic product (GDP) and human development index (HDI) (Kummu et al., 2018) were not important in the sensitivity tests probably because they were produced with sub-national data and mapped based on the same population density data as we used (Table 1). Other tested variables (e.g., livestock density, road density, forest aboveground biomass, tree cover) were excluded due to the low importance, the limited temporal coverage of data source and difficulty in extrapolation (Gilbert et al., 2018; Hansen et al., 2013; Meijer et al., 2018; Santoro et al., 2021). Sea-surface temperature has also been proved as a good indicator of El Niño-Southern Oscillation (ENSO) and fire activity, especially in the tropics (Chen et al., 2011; Fernandes et al., 2011). However, our model training and prediction are based on land grid cells, and it is difficult to incorporate the sea-surface temperature information in each land grid cell in the current framework.

In addition, sea-surface temperature is closely linked to climate variables over land through atmospheric circulation and teleconnection, thus the impacts of sea-surface temperature could have been implicitly considered in the model through the climate variables over land. Representation of anthropogenic intervention on fires (e.g., fire suppression, prescribed burns) (Libonati, 2024) cannot be fully considered by the population density used in our models due to the limited temporal coverage of related datasets. Fire suppression has been used to control fire activities for a long time in history (Douglas et al., 2001), especially in the US. 98% of wildfires were suppressed within the first 24 hours in the US (Forest Service, 2009). Meanwhile, evidence shows that conventional fire suppression could build up fuels and enlarge fire risk, thus prescribed burns are the new recognized measures for clearing accumulated fuels and relieving extreme wildfire risk (Kreider et al., 2024; Schoennagel et al., 2017). Unfortunately, fire suppression was not explicitly represented in our models due to lack of data, and it may partly explain the overestimated burned areas in continental USA and Europe during the 20th century (Fig. 9b, 9g). Moreover, landscape fragmentation, usually caused by land use and management (Driscoll et al., 2021), has also been proved to alter fire regimes (Alencar et al., 2015), fire occurrence (Silva Junior et al., 2018) and burned area trends (Rosan et al., 2022) in some regions. Nevertheless, fragmentation was not explicitly considered in the reconstruction of burned area in this study because fragmentation indices are calculated based on high-resolution land cover maps and there is no such data available before the satellite era.

Though the temporal coverage of FireCCILT11 is longer than FireCCI51, there are some known issues in FireCCILT11. For example, the orbit-drift artifacts can be many times larger in magnitude than the true burned area signal, especially in the tropics and USA, and it thus distorts burned area trends and causes inconsistency over the time span at the sub-continental scale (Giglio and Roy, 2022, 2024). We chose to use FireCCI51 for its recognized coherence and robust performance, but FireCCI51 could neglect some small fires due to its moderate spatial resolution (approximately 250 meters), even that the sensitivity to small fires in FireCC51 was improved compared with FireCCI50 (Lizundia-Loiola et al., 2020).

In this study, we tested other commonly used machine learning models in NHAF and BOAS (Sect. 2.2). In NHAF, $R^2$ between BAF observations and BAF predictions by other machine learning models is 0.54 for regular BAF and ranges from 0.29 to 0.33 for extreme BAF (Fig. S7), and LSTMs performed best with $R^2$ of 0.78 for regular BAF and 0.69 for extreme BAF (Fig. S10h, S11h). In BOAS, $R^2$ between observations and predictions by other machine learning models is 0.59-0.74 for regular BAF (Fig. S8b-S8e) and ranges from 0.36 to 0.53 for extreme BAF (Fig. S8g-S8j), and LSTMs performed best with $R^2$ of 0.75 for regular BAF and 0.56 for extreme BAF (Fig. S8a, S8f). Compared with models only using information at the current timestep, LSTMs incorporate information from previous timesteps and thus are able to include feedback effects from output to input, which is important to account for the complex interactions between fires and other factors (Hochreiter and Schmidhuber, 1997). Extrapolation studies are based on the assumption that the paradigms of interacting factors would not change in space or time, so models trained with limited data could be transferred to make predictions spatially or temporally out of the coverage of training data. However, the paradigms could be changing by space or time in the real world. Therefore, this kind of uncertainty inevitably exists spatiotemporally in the extrapolation studies.

In comparison of fire occurrence with charcoal datasets, the varying accuracy across the globe could partially be explained by the varying data quality among all sites. Meanwhile, there are large uncertainties in the age models used for calculating the age of charcoal records (Harrison et al., 2022), which affect the calculation of predicted fire occurrence accuracy to some degree. Moreover, charcoal records could not distinguish wildfires and human-induced fires, but our predictions exclude fires over croplands (Sect. 2.1), which may cause inconsistency between charcoal records and our reconstruction. The differences between our predictions and the dataset by Mouillot and Field (2005) may also be induced by some assumptions (e.g., the same trends of nearby countries if there is no historical data, prescribed fire probability map for burned area mapping) in Mouillot and Field (2005) due to lack of data and some methodology limitations. Moreover, some burned area statistics before the satellite era in some regions (e.g., US Forest Services, Food and Agriculture Organization of the United Nations) applied in Mouillot and Field (2005) could be very uncertain due to the difficulty of counting all fires across a country or state without large-scale monitoring techniques. In summary, our predictions reproduced inter-annual variability and seasonality of FireCCI51 (2003-2020) in all regions, and match well with other observation-based burned area datasets in Brazil and Chile (1985-2020) even that we fed no observed burned area before the 21st century to train our models. Our predictions are yet able to capture the spatiotemporal pattern of burned area in some regions (e.g., Australia) due to the uncertainty and missing data from data source.

## 5 Data availability

The global monthly 0.5°×0.5° burned area fraction maps from 1901 to 2020 can be freely accessed at https://doi.org/10.5281/zenodo.14191467 (Guo and Li, 2024). The availability of other datasets used in this study is noted in Table 1 and Table 2.

## 6 Conclusions

We used machine learning models to build empirical relationships of monthly burned area fraction with factors related to climate, vegetation and human activities at the 0.5°×0.5° scale. Our historical burned area product from 1901 to 2020 can be used to benchmark historical simulations for fire modules in DGVMs, re-calculate historical fire emissions and estimate legacy effects of vegetation recovery after fires on terrestrial carbon sink. Though the temporal coverage of our product is long enough to support studies related to fire disturbance, carbon dynamics and climate change, more reliable explanatory data for model training and burned area data for validation would help further improve the accuracy of the reconstructed burned area product.

## Supplementary information

The supplementary information related to this article is attached and summarized in SupplementaryInformation.docx (a Microsoft Word file).

## Author contribution

Zhixuan Guo conducted data analysis, produced the dataset and drafted the manuscript. Wei Li and Philippe Ciais proposed the idea and supervised the study. Stephen Sitch and Guido R. van der Werf provided necessary datasets for this study. Wei Li and Zhixuan Guo revised the manuscript, with Philippe Ciais, Stephen Sitch, Guido R. van der Werf, Simon P. K. Bowring, Ana Bastos, Florent Mouillot and Xiaomeng Huang contributing to methodology improvements and revision of the manuscript. Minxuan Sun, Lei Zhu, Xiaomeng Du and Nan Wang helped collect data and check the results.

## Competing interests

The contact author declared that there are no competing interests from any other authors.

## Acknowledgements

This study is funded by Yunnan Provincial Science and Technology Project at Southwest United Graduate School (grant number: 202302AO370001) and the National Key R&D Program of China (grant number: 2019YFA0606604). This work is a contribution to the CALIPSO project supported by Schmidt Sciences. This work is supported by the Center of High-Performance Computing, Tsinghua University.

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
