# Peer review of "Reconstructed global monthly burned area maps from 1901 to 2020"

_Earth System Science Data, 2024_

## Author Response (AR1)

**Response to comments**

**Manuscript ID:** essd-2024-556

**Title:** *Reconstructed global monthly burned area maps from 1901 to 2020*

**Journal:** *Earth System Science Data*

Dear editor and reviewers,
In the revision, we carefully addressed the reviewers' concerns (see point-by-point responses below) and revised the Main Text and Supplementary Information with blue for newly added and black for unchanged.

**Reviewer #1:**

**General Comments**

**Comment #1**

Earth System Science Data – Guo et al. – Feb 2025

Guo et al. have developed a novel machine learning model that can reconstruct the global monthly burned area at a spatial resolution of 0.5°×0.5° from 1901 to 2020. This model can be used to provide a benchmark for historical simulations of fire modules in Dynamic Global Vegetation Models (DGVMs). This approach employs various machine learning models to distinguish between extreme large fires and regular fires, using climate, vegetation, and human activities as explanatory variables and satellite-based burned area (FireCCI51) as the target variable to build the models. The results of model show high accuracy in some regions when compared with charcoal records. The manuscript is well-written and the results are effectively presented. In general the work is worthy publication in ESSD. I have only minor comments mainly for clarification.

**Response #1**

We thank the reviewer for the constructive and positive comments. Following your insightful suggestions, we have carefully revised the manuscript. Please find the point-by-point responses below.

**Comment #2**

Machine learning is a data-driven model, and thus the selection of data, especially the choice of explanatory variables, is crucial for model training. In this paper, variables related to climate, vegetation, and human activities were selected, and feature selection was employed to screen these variables. However, lightning, particularly cloud-to-ground lightning with sufficient energy, which is an important ignition source, was not included as an explanatory variable in

the training of the machine learning model. Additionally, large-scale climate forcing, such as sea-surface temperature, can dominate extreme fire activity and the seasonality of fire activity. I would like to know the authors' thoughts on whether these data were excluded due to insufficient temporal coverage or were screened out through feature selection.

The terrain can affect the spread of fire. However, among all the explanatory variables listed in Table 1 and Figure S8 in the article, there was no information on terrain. I guess this might be due to limitations in the availability of data?

**Response #2**

We appreciate the reviewer's great suggestion and agree that lightning and terrain information could also impact the fires. We thus added sensitivity tests by including lightning and terrain respectively as explanatory variables. Our sensitivity results showed that lightning and terrain both have minor influence on classification across all regions (Table R1). For regression, lightning and terrain both pose considerable impacts in NHAF and SHAF, with minor effects in other regions (Table R2, R3). With lightning and terrain resampling to 0.5°×0.5° grid cells, in addition to the limited time span of data, explicit information could be inevitably diluted, which partly explains the minor effect in most of regions.

Accordingly, we described these sensitivity tests in **Supplementary Information** (**Supplementary Text 3, Table R1-R3**) and in the **Section 2.2 and 4 of Main Text** as follows:

"**Supplementary Text 3**

**Sensitivity tests of including lightning and terrain as explanatory variables**

Lightning and terrain have been proved as basic and important information for understanding and predicting fire activity (Bowman et al., 2009; Bowman et al., 2020), we thus did sensitivity tests by including lightning and terrain as explanatory variables in the machine learning models.

We extracted the average of cloud-to-ground lightning power within 0.5°×0.5° grid cells for from The World Wide Lightning Location Network (WWLLN) Global Lightning Climatology (WGLC) and time series, which is a global dataset covering 2010-2024 at a spatial resolution of half-degree or 5 arc-minute (Kaplan and Lau, 2021). We then calculated the average and standard deviation of elevation, the median and standard deviation of slope gradient within 0.5°×0.5° grid cells based on Global Multiple Terrain Elevation Data (GMTED), a global dataset in 2010 at a spatial resolution up to 7.5 arc-second (Danielson and Gesch, 2011).

To save the computational cost in the sensitivity tests, we randomly selected a half of samples from 2010 to 2020 and then split it into the training set (80%) and the testing set (20%) in each region, respectively. In 0.5°×0.5° grid cells, we added the average cloud-to-ground lightning power to represent lightning, and the average and standard deviation of elevation, the median

and standard deviation of slope gradient to represent terrain as explanatory variables in addition to the original variables (Table 1) in model training and testing.

Our sensitivity results show that lightning and terrain generally manifest minor effect across all regions in classification (Table R1). For regression, lightning and terrain pose considerable impacts in NHAF and SHAF but minor effects in other regions (Table R2, R3). Lightning improved the regression model performance in NHAF for regular burned area ($R^2$ improved by ~0.1, Table R2). In contrast, terrain decreased the regression model performance the most in SHAF for regular burned area ($R^2$ decreased by ~0.05, Table R2), but it increased the regression model performance in NHAF and SHAF for extreme burned area ($R^2$ improved by ~0.05, Table R3).

Despite the importance in some regions, we did not use lightning data for historical burned area reconstruction due to the limited time span. The time span of global lightning data is 2010-2024 (Kaplan and Lau, 2021), and the historical lightning data is not accessible especially before the 21st century. Terrain resampling to 0.5°×0.5° grid cells inevitably diluted explicit information from fine spatial resolution (Cary et al., 2006), which may partly explain its minor effect across all regions except NHAF and SHAF."

**Line 167-168 in Main Text:** "In addition to the 16 explanatory variables in Table 1, we conducted sensitivity tests by incorporating lightning (Kaplan and Lau, 2021) and terrain information (Danielson and Gesch, 2011) in each region (Supplementary Text3) to assess whether these variables can help improve model performance. We also tested other variables in NHAF (e.g., gross domestic product, human development index, livestock density, road density, tree cover, forest aboveground biomass) (Fig. S8), …"

**Line 439-440 in Main Text:** "In addition to features selected in this study, other explanatory variables besides Table 2 were tested but eliminated (Fig. S8, Table R1-R3). For instance, lightning data is only available during 2010-2024, and thus it cannot be utilized to reconstruct burned area in the 20[th] century (Kaplan and Lau, 2021). Terrain resampling to 0.5°×0.5° grid cells inevitably diluted explicit information from a fine spatial resolution (Cary et al., 2006), thus posing minor effect across all regions except NHAF and SHAF. Gross domestic product (GDP) and human development index (HDI) …"

We agree that sea-surface temperature can impact extreme fire activity (e.g., seasonality). However, our model training and prediction are based on the land grid cells, so it is difficult to incorporate the sea-surface temperature information in each land grid cell in the current framework. We added sentences to clarify this point in the **Section 4 of Main Text**.

**Line 444:** "Sea-surface temperature has also been proved as a good indicator of El Niño-Southern Oscillation (ENSO) and fire activity, especially in the tropics (Chen et al., 2011;

Fernandes et al., 2011). However, our model training and prediction are based on land grid cells, and it is difficult to incorporate the sea-surface temperature information in each land grid cell in the current framework. In addition, sea-surface temperature is closely linked to climate variables over land through atmospheric circulation and teleconnection, thus the impacts of sea-surface temperature could have been implicitly considered in the model through the climate variables over land."

**Table R1 (added as Table S5 in Supplementary Information).** Evaluation of the random forest classification models using original 16 explanatory variables (Table 1) (original) and adding lightning or terrain into explanatory variables respectively in each region. Note that Area Under receiver operating characteristic Curves (AUC) ranging in 0-1, and larger AUC indicating better model performance. Colored grids represent improved (red) or decreased (blue) model performance by adding lightning and terrain as explanatory variables compared to using original explanatory variables.

| region | original | | | adding lightning | | | adding terrain | | |
|---|---|---|---|---|---|---|---|---|---|
| | overall accuracy | AUC for regular BAF | AUC for extreme BAF | overall accuracy | AUC for regular BAF | AUC for extreme BAF | overall accuracy | AUC for regular BAF | AUC for extreme BAF |
| BONA | 0.96 | 0.92 | 0.92 | 0.96 | 0.93 | 0.95 | 0.97 | 0.93 | 0.92 |
| TENA | 0.91 | 0.84 | 0.92 | 0.90 | 0.83 | 0.90 | 0.91 | 0.86 | 0.93 |
| CEAM | 0.90 | 0.89 | 0.95 | 0.90 | 0.89 | 0.94 | 0.89 | 0.89 | 0.95 |
| NHSA | 0.90 | 0.92 | 0.99 | 0.90 | 0.93 | 0.99 | 0.90 | 0.93 | 0.99 |
| SHSA | 0.87 | 0.89 | 0.98 | 0.87 | 0.89 | 0.97 | 0.88 | 0.90 | 0.98 |
| EURO | 0.98 | 0.95 | 0.96 | 0.98 | 0.95 | 0.95 | 0.98 | 0.96 | 0.97 |
| MIDE | 0.98 | 0.97 | 0.97 | 0.98 | 0.97 | 0.95 | 0.98 | 0.97 | 0.95 |
| NHAF | 0.90 | 0.96 | 0.99 | 0.89 | 0.95 | 0.99 | 0.90 | 0.96 | 0.99 |
| SHAF | 0.88 | 0.94 | 0.98 | 0.88 | 0.94 | 0.98 | 0.88 | 0.94 | 0.98 |
| BOAS | 0.95 | 0.93 | 0.96 | 0.95 | 0.94 | 0.96 | 0.96 | 0.94 | 0.96 |
| CEAS | 0.94 | 0.92 | 0.96 | 0.94 | 0.92 | 0.96 | 0.94 | 0.93 | 0.96 |
| SEAS | 0.92 | 0.93 | 0.98 | 0.92 | 0.93 | 0.98 | 0.93 | 0.94 | 0.98 |
| EQAS | 0.91 | 0.88 | 0.95 | 0.92 | 0.88 | 0.94 | 0.90 | 0.88 | 0.95 |
| AUST | 0.87 | 0.88 | 0.95 | 0.87 | 0.88 | 0.95 | 0.88 | 0.88 | 0.95 |

**Table R2 (added as Table S6 in Supplementary Information).** Evaluation of the long short-term memory (LSTM) regression models using original 16 explanatory variables (Table 1) (original) and adding lightning or terrain into explanatory variables respectively for regular burned area in each region. $R^2$, slope, and RMSE represents the coefficients of determination, linear slope and rooted mean squared error between prediction and observation in the testing

set. Colored grids represent improved (red) or decreased (blue) model performance by adding lightning or terrain as explanatory variables compared to using original explanatory variables.

| Region | original | | | adding lightning | | | adding terrain | | |
|--------|----------|-------|------|------------------|-------|------|----------------|-------|------|
| | $R^2$ | slope | RMSE | $R^2$ | slope | RMSE | $R^2$ | slope | RMSE |
| BONA | 0.85 | 0.44 | 0.004 | 0.86 | 0.46 | 0.003 | 0.85 | 0.42 | 0.005 |
| TENA | 0.92 | 0.43 | 0.001 | 0.91 | 0.42 | 0.002 | 0.92 | 0.44 | 0.001 |
| CEAM | 0.64 | 0.43 | 0.002 | 0.65 | 0.47 | 0.002 | 0.66 | 0.48 | 0.002 |
| NHSA | 0.66 | 0.45 | 0.004 | 0.66 | 0.45 | 0.004 | 0.68 | 0.45 | 0.004 |
| SHSA | 0.83 | 0.50 | 0.002 | 0.83 | 0.50 | 0.002 | 0.84 | 0.55 | 0.002 |
| EURO | 0.91 | 0.45 | 0.001 | 0.91 | 0.46 | 0.001 | 0.91 | 0.47 | 0.001 |
| MIDE | 0.86 | 0.43 | 0.001 | 0.86 | 0.43 | 0.001 | 0.87 | 0.43 | 0.001 |
| NHAF | 0.77 | 0.58 | 0.019 | 0.87 | 0.68 | 0.011 | 0.75 | 0.55 | 0.019 |
| SHAF | 0.77 | 0.62 | 0.017 | 0.80 | 0.64 | 0.015 | 0.72 | 0.55 | 0.018 |
| BOAS | 0.65 | 0.33 | 0.005 | 0.65 | 0.34 | 0.005 | 0.67 | 0.35 | 0.004 |
| CEAS | 0.73 | 0.30 | 0.002 | 0.72 | 0.30 | 0.002 | 0.73 | 0.32 | 0.002 |
| SEAS | 0.71 | 0.35 | 0.004 | 0.72 | 0.35 | 0.004 | 0.73 | 0.35 | 0.004 |
| EQAS | 0.98 | 0.41 | 0.001 | 0.98 | 0.43 | 0.001 | 0.98 | 0.44 | 0.001 |
| AUST | 0.80 | 0.45 | 0.011 | 0.81 | 0.45 | 0.010 | 0.81 | 0.45 | 0.010 |

**Table R3 (added as Table S7).** Same as Table R2 but for extreme burned area.

| Region | original | | | adding lightning | | | adding terrain | | |
|--------|----------|-------|------|------------------|-------|------|----------------|-------|------|
| | $R^2$ | slope | RMSE | $R^2$ | slope | RMSE | $R^2$ | slope | RMSE |
| BONA | 0.86 | 0.47 | 0.07 | 0.88 | 0.54 | 0.05 | 0.86 | 0.52 | 0.07 |
| TENA | 0.87 | 0.46 | 0.03 | 0.87 | 0.46 | 0.03 | 0.87 | 0.51 | 0.03 |
| CEAM | 0.96 | 0.41 | 0.02 | 0.96 | 0.41 | 0.02 | 0.96 | 0.45 | 0.02 |
| NHSA | 0.67 | 0.39 | 0.03 | 0.67 | 0.41 | 0.03 | 0.67 | 0.39 | 0.03 |
| SHSA | 0.78 | 0.41 | 0.06 | 0.78 | 0.43 | 0.06 | 0.78 | 0.41 | 0.06 |
| EURO | 1.00 | 0.40 | 0.01 | 1.00 | 0.40 | 0.01 | 1.00 | 0.40 | 0.01 |
| MIDE | 0.96 | 0.42 | 0.01 | 0.96 | 0.42 | 0.01 | 0.96 | 0.42 | 0.01 |
| NHAF | 0.65 | 0.51 | 0.16 | 0.67 | 0.51 | 0.15 | 0.70 | 0.55 | 0.15 |
| SHAF | 0.65 | 0.46 | 0.13 | 0.62 | 0.42 | 0.14 | 0.70 | 0.47 | 0.12 |
| BOAS | 0.58 | 0.44 | 0.08 | 0.59 | 0.51 | 0.07 | 0.60 | 0.50 | 0.08 |
| CEAS | 0.86 | 0.46 | 0.06 | 0.86 | 0.44 | 0.06 | 0.87 | 0.46 | 0.06 |
| SEAS | 0.68 | 0.48 | 0.06 | 0.68 | 0.48 | 0.06 | 0.60 | 0.45 | 0.07 |
| EQAS | 0.88 | 0.40 | 0.01 | 0.88 | 0.37 | 0.01 | 0.89 | 0.40 | 0.01 |
| AUST | 0.67 | 0.43 | 0.13 | 0.69 | 0.45 | 0.12 | 0.68 | 0.46 | 0.13 |

**Comment #4**

It is unclear for me whether you do training-validation separately for each GFED region or you just do training-validation for NHAF and then apply the model globally? I.e., did you build a

single model by using NHAF followed by its application everywhere? Or each region has its own model?

**Response #4**

We conducted model training, testing and prediction for each region individually, and we added further clarification in the **Section 2 of Main Text** as follows:

**Line 86-88:** "We first divided the globe into 14 regions (Fig. S1) following the Global Fire Emission Dataset (GFED regions) (Giglio et al., 2006; Van Der Werf et al., 2017) and conducted machine learning model training, testing and prediction in each GFED region individually."

**Line 147-148:** "For each region (Fig. S1), we fed BAF as the dependent variable, and the 16 explanatory variables (Table 1) as independent variables to build the machine learning models individually."

**Comment #5**

Why not consider using different types of ML model for different GFED regions if we can pick a best type of model for each region? Is this because of computation resource limitation?

**Response #5**

Thanks for the reviewer's comment. We tested several machine learning models in NHAF in the original version of manuscript because NHAF has almost the largest burned area among all GFED regions. It turned out LSTMs exhibited higher performance than other tested models. LSTMs require the time series of input data and are able to integrate information over long time series, which is an advantage for understanding fire activity comparing with other machine learning models that don't integrate information from previous time steps.

We clarified this point on **Line 158-165** in **Main Text**: "We chose NHAF as the testing region because its annual total burned area dominates global annual total burned area, and our preliminary tests severely underestimated the annual total burned area in NHAF, thus we aimed to improve model performance in NHAF by testing different machine learning models. In this test, we took only one year data (2010) and split it into the training set (80%) and the testing set (20%). It turned out that LSTMs have the best performance (Fig. 2k, S7) for regression with a memory window of three months. LSTMs consist of three gated memory cells (input gate, forget gate, output gate) that can integrate information over long time series (Hochreiter and Schmidhuber, 1997), exhibiting good performance on extreme events (e.g., precipitation, floods) (De Sousa Araújo et al., 2022; Nearing et al., 2024)."

In addition to the tests in NHAF, we also conducted additional tests of multiple machine learning models in BOAS (boreal Asia), which is another representative region for fires, in this

revision. We added description in the **Section 2.2 and Section 4 of Main Text**, and added **Fig. R1** in **Supplementary Information** as follows:

**Line 155-165 in Main Text:** "For the regression model selection, we tested commonly used machine learning models including random forest (Tin Kam, 1995), quantile random forest (Meinshausen, 2006), gradient boosting (Friedman, 2001) and extreme gradient boosting (Chen and Guestrin, 2016), and a deep learning architecture called long short-term memory networks (LSTMs) (Hochreiter and Schmidhuber, 1997) in NHAF and BOAS. We chose NHAF as the testing region because its annual total burned area dominates global annual total burned area, and our preliminary tests severely underestimated the annual total burned area in NHAF, thus we aimed to improve model performance in NHAF by testing different machine learning models. In addition, we selected BOAS as another testing region because this region experiences regular fires but has different climate and vegetation conditions from NHAF. In this test, we took only one year data (2010) and split it into the training set (80%) and the testing set (20%). It also shows that LSTMs have the best performance (Fig. 2k, S7, R1) for regression with a memory window of three months. LSTMs consist of three gated memory cells (input gate, forget gate, output gate) that require the time series of input data and can integrate information over long time series (Hochreiter and Schmidhuber, 1997), exhibiting good performance on extreme events (e.g., precipitation, floods) (De Sousa Araújo et al., 2022; Nearing et al., 2024)."

[Figure]

**Figure R1 (added as Fig. S7 in Supplementary Information).** Same as Fig. S7 but in BOAS. Evaluation of different machine learning regression models for regular (a-e) and extreme BAF (f-j), respectively using the testing set (20%) in 2010 in BOAS.

**Line 464 in Main Text:** "In this study, we tested other commonly used machine learning models in NHAF and BOAS (Sect. 2.2). In NHAF, $R^2$ between BAF observations and BAF predictions…"

**Line 465 in Main Text:** "In BOAS, $R^2$ between observations and predictions by other machine learning models is 0.59-0.74 for regular BAF (Fig. R1b-R1e) and ranges from 0.36 to 0.53 for extreme BAF (Fig. R1g-R1j), and LSTMs performed best with $R^2$ of 0.75 for regular BAF and 0.56 for extreme BAF (Fig. R1a, R1f)."

**Comment #6**

I understand that two types of ML models were built: classification model for extreme fire grid cell and regression model to predict BAF. My question is, if a certain grid cell was classified as an extreme fire grid cell, how its BAF was determined? The grid cells with greater than 90th quantile was classified as extreme fire but still, we need to know its specific BAF?

**Response #6**

If a certain grid cell was classified as an extreme fire grid cell, the specific BAF was further estimated by the regression model. To clarify these concerns, we added explanations in the **Section 2.2 of Main Text** as follows:

**Line 153-154:** "After classification, then we perform machine learning regressions separately for grid cells with regular or extreme BAF, and grid cells for each category (regular and extreme) are fed into separate regression models to estimate the specific BAF value (continuous values)."

**Comment #7**

Although fire is prevalent, I guess there are many zero-BAF grid cells compared with relatively small number of grid cells with BAF>0? Did you encounter the issue of imbalanced sample size? Like there are many pixels without fire but only a small fraction with fire, will this have an impact on the model building?

**Response #7**

We clarified the issue of imbalanced sample size in the **Section 2.2 of Main Text** as follows:

**Line 154:** "To balance sample sizes across BAF types, we applied a weighting method in machine learning classification models. Let the sample counts for no BAF, regular BAF, and extreme BAF be $n_1$, $n_2$, and $n_3$, respectively. We computed their least common multiple, M, and assigned weights of $M/n_1$, $M/n_2$, and $M/n_3$ to each BAF type."

**Specific Comments**

**Comment #8**

Line 172-175: could the authors give more details on how model parameter optimization was made and which parameters have been optimized? This part is interesting. Is it a 5-fold cross validation or a circular process?

**Response #8**

As suggested, we added details about the model parameter optimization in the **Section 2.2 of Main Text** as follows:

**Line 172-173:** "We randomly split the data over the period of 2003-2020 into five folds, using one fold (20%) as the testing set and the remaining four folds as the training set (80%). This process was looped for each of the five folds."

**Line 175:** "We optimized model hyperparameters using a grid search with five-fold cross-validation. For the Random Forest classifiers, we tuned "max_depth" and "n_estimators"; for the LSTM regressors, we tuned "hidden_sizes", "learning_rate", and "epochs". All combinations of these parameter values were used to retrain the models, and performance was evaluated on each held-out fold using coefficient of determination, slope, and rooted mean squared error. The combination yielding the best average metrics across folds was selected as optimal."

**Comment #9**

Line 176: After the 80%-20% 5-fold CV, here I think we call it 'model evaluation', which is better than 'model validation'. We cannot really 'validate' a model.

**Response #9**

As suggested, we've revised 'model validation' to 'model evaluation' thoroughly in **Main Text** (including text in Fig. 1) and **Supplementary Information**.

**Comment #10**

Line 183-184: here is confusing. I get confused by whether the model was validated by 80%-20% as in line 172-175 or by leave-one-year-out?

**Response #10**

We revised the sentences to make it clear **in Main Text**.

**Line 176:** "After determining the optimal model parameters, we conducted model evaluation using a leave-one-year-out method in addition to the 5-fold evaluation method in the model parameterization process."

**Line 183-184:** "The machine learning models with optimal parameters from the 5-fold evaluation process were finally used to predict global monthly BAF maps from 1901 to 2020."

**Comment #11**

On lines 161–162 of the manuscript, the authors stated that LSTMs have the best performance among all the machine learning models used in this study, and this result was presented in Fig. 2k and Fig. S7. As far as I understand, Fig. 2 was intended to display the results of LSTMs, but the description in the figure title was unclear. I would like to see a clearer statement and further explanation of what the "absolute and relative difference" mean in Fig. 2b.

**Response #11**

As suggested, we clarified these issues in the revised manuscript version. Please note that Fig. 2 in the original version **was split into Fig. R2 and Fig. R3** in the revised version based on the comments from **Reviewer #2**. In addition, we also removed dots with both predicted and observed burned area fraction equal to 0 in the scatter plots of Fig. 2 because we aimed to emphasize the evaluation of model performance among grid cells with BAF>0. Therefore, N, $R^2$, slope, p and RMSE denoted in scatter plots could be different from Fig. 2 in the original version. We clarified this change in the caption **in Main Text**.

[revised manuscript text omitted]

**Response to comments**

**Manuscript ID:** essd-2024-556

**Title:** *Reconstructed global monthly burned area maps from 1901 to 2020*

**Journal:** *Earth System Science Data*

Dear editor and reviewers,
In the revision, we carefully addressed the reviewers' concerns (see point-by-point responses below) and revised the Main Text and Supplementary Information with blue for newly added and black for unchanged.

**Reviewer #2:**

**General Comments**

**Comment #1**

Earth System Science Data – Guo et al. – Feb 2025
This study reconstructed global monthly burned area from 1901 to 2020 at 0.5°×0.5° resolution using machine learning models trained on satellite data (2003–2020). Separate models were developed for extreme and regular fires based on climate, vegetation, and human activity data. The models accurately captured spatial patterns, seasonal trends, and long-term changes in burned area. Results show a global decline in burned area from 1901–1978, an increase from 1978–2008, and a stronger decline from 2008–2020. The reconstruction aligns well with charcoal records, offering a valuable tool for historical fire analysis.

The manuscript is well written, and the simulation work—including data preparation, model selection, training, validation, and regional application—is rigorously conducted. It can be shown that the authors put quite a lot effort to build such machine learning and data driven pipeline. The use cases from such data asset construction are wide. The results are clearly presented through both text and figures, and the discussion addresses the burned area distributions, trends, limitations, and uncertainties. However, I have a few specific comments regarding the methods that I believe should be clarified in more detail in the Methods section. Therefore, I recommend a minor revision to further strengthen the manuscript prior to publication.

**Response #1**

We thank the reviewer for the constructive and positive comments. Following your insightful suggestions, we have carefully revised the manuscript. Please find the point-by-point responses below.

**Specific Comments**

**Comment #2**

Line 95 Please add more content / citations on how the burned area on cropland is excluded to eliminate agricultural fires in this study.

**Response #2**

As suggested, we revised the description in the **Section 2.1 of Main Text**.

**Line 95-96:** "We excluded all burned pixels overlapping cropland classes in the CCI land-cover layer provided with FireCCI51 (Lizundia-Loiola et al., 2020) to remove agricultural fires from our analysis."

**Comment #3**

Line 120 Please clarify how the reclassification is done to split them into five land use types (forest, shrub, natural grass, cropland and others). Was it via machine learning classifier?

**Response #3**

As suggested, we added description accordingly in the **Section 2.1 of Main Text**.

**Line 121:** "The above five land use types were converted from the ESA CCI land cover maps based on the cross-walking table (Li et al., 2018). For the LUH2 dataset, we reclassified land use by summing forested primary land (primf) and potentially forested secondary land (secdf) to create a single "forest" category, and by summing all crop types (c3ann, c3per, c3nfx, c4ann, and c4per) to form the "cropland" category. To define natural grass and shrub, we first combined non-forested primary land (primn) and potentially non-forested secondary land (secdn) into a unified grass + shrub type. We then allocated this combined area back into separate grass and shrub categories based on their proportional distribution. For historical years before 1992, the proportional distribution was set the same as ESA CCI land cover in 1992, and for years in 1992-2020, the proportional distribution was set according to the corresponding year of ESA CCI land cover."

**Comment #4**

Line 148 Is there reasoning why an initial classification has to be conducted to classify regions into non-BAF, moderate and extreme BAF? Why couldn't the parameterization be applied to all regions without this initial classification. Conducting such classification could introduce extra errors / uncertainties. Please add more content to clarify.

**Response #4**

Thanks for the reviewer's comment. We need this initial classification because of imbalanced samples. We explained it in **Line 96-98 in Main Text**: "The distribution of monthly burned area within half-degree grid cells manifests that split of regular and extreme burned area amplifies the kernel density of extreme burned area for better model training (Fig. S2)."

Due to the imbalanced samples, too many grid cells with no BAF and regular BAF would lower the weights of extreme BAF in the model training, leading to underestimation of the large BAF and the total burned area. We thus excluded zero-BAF and conducted separate regression models for regular and extreme BAF. We added further clarification in the **Section 2.1 of Main Text** as follows:

**Line 96-98:** "The monthly distribution of burned area within 0.5°×0.5° grid cells (Fig. S2) shows that if regular and extreme fires are modeled together (black curves), the abundant moderate values drown out the extremes (orange curves), causing total area to be underestimated. We thus first conducted classification and then trained separate models for regular and extreme burned fractions to enhance the representation of extreme events and improve regression performance."

**Comment #5**

Line 152 I am not quite clear why 90th percentile instead of other percentiles is used to determine the extreme BAF. Please explain and clarify.

**Response #5**

As suggested, we added further clarification in the **Section 2.1 of Main Text** to explained why $90^{th}$ percentile was chosen in the classification.

**Line 96:** "We used the $90^{th}$ percentile of all burned area fractions in 0.5°×0.5° grid cells within a region as the threshold to define extreme fires. This percentile was chosen based on previous literature (Bowman et al., 2017; Cunningham et al., 2024; Lannom et al., 2014). It is high enough ($\geq 90^{th}$) to distinguish moderate from extreme samples to train separate models for each category. Meanwhile, it is not too high (e.g., $95^{th}$ or $99^{th}$) in regions with limited data (such as Europe and the Middle East) to ensure sufficient extreme samples for model training and evaluation."

**Comment #6**

Line 167 From the content there are only 16 features plus some others in NHAF are used in the model. I am not sure why a recursive feature selection has to be performed. Recursive feature selection is usually applied when there is 1000+ variables used in GBM models. This process is necessary because the larger model package size would directly cause latency issues in the

live production env. In this case 16+ variables won't cause such issues plue the model will not be deployed onto live env. Is such feature selection necessary?

**Response #6**

We agree that recursive feature selection is essential when working with a small sample size and many candidate predictors. Here, we used it to prevent performance degradation that can occur when adding irrelevant features (Guyon and Elisseeff, 2003). Moreover, reducing the feature set enhances interpretability—facilitating the analysis of feature interactions via SHAP—and conserves computational resources. We added explanation in the **Section 2.2 of Main Text** as follows:

**Line 170:** "The recursive feature elimination cross validation was applied to prevent model performance degradation if irrelevant features were added (Guyon and Elisseeff, 2003). Moreover, reducing the feature set could enhance model interpretability and conserve computational resources (Lundberg et al., 2020)."

**Comment #7**

Figure 2 For the global map of burned area difference please make it a separate figure as it is hard to see the small dots. And what is the unit? Is it a relevant difference? Please specify in the figure.

**Response #7**

As suggested, we split Figure 2 of the previous version into Figure R2 and Figure R3. The unit of the global map in Fig. R2 is fraction difference in 0.5°×0.5° grid cell, and it is not a relative difference. We clarified it in the caption of Fig. R2 **in Main Text**.

**Comment #8**

Figure 2 What does N mean here? Is it the data sample size for validation grids per year? Please specify.

**Response #8**

N represents number of grid cells with multi-year averaged BAF>0. We removed dots with both predicted and observed burned area fraction equal to 0 in the scatter plots of Fig. 2 because we aimed to emphasize the evaluation of model performance among grid cells with BAF>0. Therefore, N, R2, slope, p and RMSE denoted in scatter plots could be different from Fig. 2 in the original version. We clarified it in the caption (now **Fig. 3 in Main Text**).

[Figure]

[Figure]

**Figure R2 (revised as Figure 2 in Main Text):** Multi-year (2003-2020) averaged burned area difference between our predictions by the leave-one-year-out method and FireCCI51 observations (predictions minus observations). (a) Map of burned area fraction difference in each 0.5°×0.5° grid cell. Burned area fraction difference is the ratio of burned area difference to total grid area within each 0.5°×0.5° cell, making it unitless and bounded between 0 and 1. (b) Latitudal sum of burned area difference using the burned area fraction difference map from (a) multiplied by the area of each 0.5°×0.5° grid cell. Both absolute (solid line) and relative (dashed line) differences are shown.

[Figure]

**Figure R3 (revised as Figure 3 in Main Text):** Scatter plots of multi-year (2003-2020) averaged burned area fraction (BAF) in each 0.5°×0.5° grid cell from predictions by the leave-one-year-out method and FireCCI51 observations for each region (a-o). Dots represent grid cells with BAF>0 averaged over 2003-2020. N, $R^2$, slope, p and RMSE respectively represent number of grid cells with multi-year averaged BAF>0, coefficient of determination, linear slope, p-value for linear correlation and rooted mean squared error between BAF from our predictions and observations. Burned area fraction is the ratio of burned area to total grid area within each 0.5°×0.5° cell, making it unitless and bounded between 0 and 1.
* * *
**Comment #9**

Figure 3 It looks like figure (a) (c) (e) vs figure (b) (d) (f) are giving the same messages. Also it is hard to find any insight from figure (a) (c) (e). Consider removing or merging them with (b) (d) (f).

**Response #9**

As suggested, we merge (b), (d), (f) into (a), (c), (e) as shown in **Fig. R4**. We revised the caption of this figure **in Main Text** accordingly as follows.

[Figure]

**Figure R4 (revised as Figure 4 in Main Text):** Mean absolute SHAP value and the ranking of all input variables (Table 1) using the random forest classification models (a) and the LSTMs regression models for regular (b) and extreme (c) BAF, respectively, in each region. Numbers denoted in grids are the ranking of variables, and higher ranking represents relative higher mean absolute SHAP value in the corresponding GFED region.

**Comment #10**

Figure 5 It is hard to tell the accuracy difference in (a) (b) as most of them are dark red (are most of them 100% or 70%?). Please consider using a different color scale to make them more distinguishable.

**Response #10**

We changed the color scale as suggested (**Fig. R5**) **in Main Text**.

[Figure]

**Figure R5 (revised as Figure 6 in Main Text):** Fire occurrence comparison between two charcoal record databases and our prediction from 1901 to 2020. (a) Site accuracy map using Global Charcoal Database. The site accuracy (%) is equal to the number of records with predicted burned area dividing the number of all records multiplying by 100%. (b) The same as (a) but using Reading Palaeofire Database instead. (c) Accuracy time series using Global

Charcoal Database and Reading Palaeofire Database respectively. Note that only records with record year ± record age uncertainty overlapping with 1901-2020 are taken into consideration.

**Comment #11**

Figure 6 Again, (a) (b) (c) convey important messages of spatial distribution of burned areas. but they are too small in terms of figure size. Please make them bigger / clearer.

**Response #11**

We split the **Fig. 6** in the previous version into **Fig. R6** and **Fig. R7** in the current version to make them bigger and clearer. We also revised their captions and citations accordingly **in Main Text** as follows:

[Figure]

**Figure R6 (revised as Figure 7 in Main Text):** Maps of burned area fraction difference between our predictions and other global burned area datasets. (a) Map of multi-year average (1982-2018) burned area fraction difference map between our predictions and FireCCILT11 (the former minus the latter). (b, c) Same as (a) but using Global Annual Burned Area Maps (GABAM) (1985-2020) and Mouillot and Field (2005) (1901-1999) instead respectively. Note that there are several years (1986, 1988, 1990, 1991, 1993, 1994, 1997 and 1999) without available data before 2000 in GABAM.

[Figure]

**Figure R7 (revised as Figure 8 in Main Text):** Time series of annual total burned area across the globe (a) and in each region (b-o) from our predictions (red lines), Mouillot and Field (2005) (blue lines), FireCCILT11 (grey lines) and GABAM (purple lines). The breakpoints and significant slopes (p-value<0.05) were calculated by methods mentioned in Sect. 2.2. Note that there are several years (1986, 1988, 1990, 1991, 1993, 1994, 1997 and 1999) without available data before 2000 in GABAM, and thus breakpoint detection and linear slopes were applied after 2000 for this dataset.